# Antibody prophylaxis may mask subclinical SIV infections in macaques

Christopher A. Gonelli[1], Hannah A. D. King[1,2,3], SungYoul Ko[1], Christine M. Fennessey[4], Nami Iwamoto[1], Rosemarie D. Mason[1], Ashley Heimann[1], Dillon R. Flebbe[1], John-Paul Todd[1], Kathryn E. Foulds[1], Brandon F. Keele[4], Jeffrey D. Lifson[4], Richard A. Koup[1] & Mario Roederer[1✉]

Broadly neutralizing antibodies (bNAbs) show potential to prevent human immunodeficiency virus (HIV-1) infection in humans[1]. However, there are limited data on the antibody concentrations required to prevent infection. Clinical trials of bNAb prophylaxis have demonstrated partial efficacy[2], but the sampling frequency typically does not allow precise timing of infection events and concurrent antibody levels. Here, using simian immunodeficiency virus (SIV) infection of rhesus macaques, we show that although potent bNAbs can delay the onset of acute viremia, subclinical infections occur while bNAb levels remain high. Serial SIV challenge of monkeys given partially and fully neutralizing bNAbs revealed that 'viral blips'—low and transient plasma viremia—often occur while serum bNAb concentrations are well above currently accepted protective levels. To understand the precise timing of the infections resulting in such blips, we performed plasma viral sequencing on monkeys that were serially challenged with genetically barcoded SIV after bNAb administration. These analyses showed that subclinical infections occurred in most animals that were given potent bNAb prophylaxis. These subclinical infections occurred while antibody concentrations were 2- to 400-fold higher than the levels required to prevent fully viremic breakthrough infection. This study demonstrates that immunoprophylaxis can mask subclinical infections, which may affect the interpretation of prophylactic HIV-1 bNAb clinical trials.

Human immunodeficiency virus (HIV-1) was discovered more than four decades ago and continues to spread, with 1.3 million new infections globally in 2022 (ref. 3). In the absence of an effective vaccine, antiretroviral pre-exposure prophylaxis and behavioural measures aimed at preventing HIV acquisition have helped to limit HIV infections[4,5]; however, longer-lasting antiviral agents are required for more effective prevention. An appealing approach to prevent virus infection is the administration of broadly neutralizing antibodies (bNAbs) against HIV-1, given that these are generally well tolerated when injected and can persist long term in the circulation[6]. Human bNAbs are isolated from people living with HIV with potent plasma neutralizing activity against diverse strains of HIV-1 and target relatively conserved epitopes of the virion surface-expressed envelope glycoprotein (Env)[7–9].

Many different bNAbs targeting several Env epitopes including the CD4 binding site (CD4bs)[10–17], variable region 2 (V2) loop[15,18] and base of the V3 loop[15–17,19,20] have been assessed for safety and tolerability in the clinic, either alone or in combinations of two or more antibodies. Antibodies including CAP256V2LS (V2 loop) and VRC07-523LS (CD4bs) are being assessed in continuing immunoprophylactic trials[21]. The only bNAb to have undergone evaluation of its prophylactic efficacy is the CD4bs-specific VRC01; this evaluation took place as part of the Antibody Mediated Prevention (AMP) trials. These studies enrolled individuals with greater likelihood of HIV acquisition who received up to ten intravenous infusions of VRC01 monoclonal antibody (mAb) every 8 weeks, with HIV-1 diagnostic tests performed every 4 weeks[22]. Although the trials did not demonstrate efficacy against all viruses, VRC01 showed persistent 75% prevention against isolates of HIV-1 that were sensitive at an in vitro neutralization 80% inhibitory concentration ($IC_{80}$) of less than 1.0 μg ml$^{-1}$ (ref. 2). Subsequent analysis of AMP trial data showed that the serum concentration of bNAb relative to the $IC_{80}$ was correlated with prevention efficacy, with plasma bNAb levels approximately 200-fold greater than the $IC_{80}$ associated with 90% prevention efficacy[23]. These protective bNAb levels were similar, albeit 2.3-fold higher, to those required for the same protective efficacy in a meta-analysis of non-human primate (NHP) high-dose challenge model studies using a simian/human immunodeficiency virus (SHIV) challenge[24].

In the present study, we aimed to better characterize the levels of plasma bNAb at the time of breakthrough infection using an authentic NHP and simian immunodeficiency virus (SIV) challenge model. SIV challenge in rhesus macaques is a robust preclinical model[25], especially when used in combination with SIV-specific bNAbs. The only previous SIV monoclonal antibody prevention study used low-potency first-generation mAbs[26] expressed from viral vectors, yielding lower levels of plasma mAb than can be achieved through passive infusion, and used a tier 1 SIV challenge[27]. Here, passive administration of these

[1]Vaccine Research Center, National Institute of Allergy and Infectious Diseases, National Institutes of Health, Bethesda, MD, USA. [2]U.S. Military HIV Research Program, Walter Reed Army Institute of Research, Silver Spring, MD, USA. [3]Henry M. Jackson Foundation for the Advancement of Military Medicine, Bethesda, MD, USA. [4]AIDS and Cancer Virus Program, Frederick National Laboratory for Cancer Research, Frederick, MD, USA. ✉e-mail: marior@mail.nih.gov

bNAbs, as well as more recent higher-potency SIV bNAbs[28], was performed before repeated tier 2 or tier 3 SIV challenges, and infection time was correlated with bNAb levels. Barcoded SIV stocks were used to accurately define the exact time of infection.

## Fully neutralizing mAbs delay infection

A study was designed to determine the amount of infused neutralizing mAbs alone or in combination required to prevent SIV infection in NHPs. The mAbs were specific for either the SIV Env V1 (ITS06.02) or the SIV Env CD4 binding site (CD4bs) (ITS01 and ITS103.01). They also differed in their ability to completely neutralize tier 2 SIVsmE660 virus, with ITS01 and ITS06.02 reaching only approximately 55% and 30% maximal neutralization, respectively; thus, $IC_{80}$ titres could not be calculated (Extended Data Fig. 1a and Supplementary Table 1). By contrast, ITS103.01 achieved complete neutralization with an in vitro neutralization $IC_{80}$ of 0.052 µg ml$^{-1}$ and a nine-fold higher in vitro neutralization 50% inhibitory concentration ($IC_{50}$) compared with ITS01 (0.017 versus 0.154 µg ml$^{-1}$). Therefore, ITS01 and ITS06.02 were considered to be partially neutralizing, whereas ITS103.01 was a fully neutralizing antibody. Groups of six animals each were infused with the mAb ITS06.02 or the mAb ITS103.01 alone (groups 2 and 3, respectively), or with a combination of mAbs ITS01 and ITS06.02 (group 4, both partially neutralizing) or mAbs ITS103.01 and ITS06.02 (group 5, codelivery of partially and fully neutralizing mAbs). A final group remained untreated as a control (group 1) (Fig. 1a). Five days later, to allow full mucosal equilibration of the infused mAbs, weekly intrarectal challenges of SIVsmE660.FL14-IAKN (a tier 2 clonal virus) were initiated and continued until sustained viremia occurred in each animal.

The control group animals reached peak viral load within 11–25 days following the first challenge (Fig. 1b). Similarly, animals treated with ITS06.02 or ITS01 + ITS06.02 reached peak viral load between 11 and 28 days or 11 and 49 days, respectively, after the first challenge. Animals receiving ITS103.01 or ITS103.01 + ITS06.02 showed delayed onset of viremia, with their respective peak viral loads occurring at 67–151 or 46–158 days from the first viral challenge. mAb treatment did not affect set-point viremia; geometric mean viral loads from days 45–120 after the viral load first reached 1,000 copies per millilitre of plasma were comparable among groups 1–5 (median log$_{10}$ values of 4.40, 4.72, 4.41, 5.74 and 4.19, respectively) (Fig. 1c). The date of likely infection was estimated to be the challenge immediately preceding the occurrence of sustained viremia (that is, viremia that persisted, with the viral load reaching 1,000 copies per millilitre). Comparison of survival curves for time to likely infection showed no difference between control animals and those treated with ITS06.02 or ITS01 + ITS06.02 (Fig. 1d). However, groups receiving ITS103.01 alone or ITS103.01 + ITS06.02 both demonstrated significantly delayed time to infection (each $P = 0.007$). Given that mAb ITS103.01 was common to these groups, we conclude that fully neutralizing activity is required for efficacy. In an ad hoc analysis, we further combined the ITS06.02 and ITS01 + ITS06.02 animals into a '−ITS103.01 mAb' group and the ITS103.01 and ITS103.01 + ITS06.02 animals into a '+ITS103.01 mAb' group. Plotting the time to infection for these combined groups further demonstrated that only animals receiving mAbs capable of full neutralization exhibited significant delay in time to infection ($P = 6.5 \times 10^{-5}$) (Fig. 1e).

## Infection can occur at high mAb levels

The kinetics of viral replication did not markedly differ between control animals and those receiving fully or partially neutralizing mAbs, as evidenced by their being no change in peak plasma viral load (Fig. 2a) or the rate of viremia onset as characterized by the upslope when synchronizing viral loads at 1,000 copies per millilitre of plasma (Fig. 2b). However, animals infused with the fully neutralizing ITS103.01 mAb demonstrated a reduced viremia upslope versus animals receiving only

partially neutralizing mAbs ($P = 0.028$), suggesting that waning levels of ITS103.01 may have blunted early viral replication. The lack of difference in upslope compared with that of the control group was probably due to the smaller control group size ($n = 6$) relative to the larger treatment group sizes (each $n = 12$), because animals were pooled according to ITS103.01 administration. Notably, half of the ITS103.01-treated animals exhibited viral 'blips' (transient viral loads of less than 1,000 copies per millilitre of plasma followed by one or more time points of undetectable viral load) in the time between the first virus challenge and the occurrence of sustained viremia (Extended Data Fig. 2).

To identify the plasma levels of mAbs necessary for protection, the pharmacokinetics of mAbs were assessed in all animals (Fig. 2d). This analysis showed similar median plasma half-lives of 12.8 and 12.6 days for ITS01 and ITS06.02, respectively, whereas that of ITS103.01 was shorter at 7.8 days (Fig. 2c). The half-lives of these mAbs were not affected by whether they were administered alone or in combination with another mAb. Analysis of mAb levels at the time of virus infection and/or replication focused on the fully neutralizing mAb ITS103.01 (in animals from groups 3 and 5), as only administration of ITS103.01 significantly affected the time to virus infection. The median plasma ITS103.01 concentration at the time of the challenge likely to have caused infection was 0.74 µg ml$^{-1}$, 14.2-fold greater than the in vitro neutralization $IC_{80}$ of 0.052 µg ml$^{-1}$ (Fig. 2e). Once acute viremia had occurred (using the time when the viral load reached 1,000 copies per millilitre of plasma), mAb levels were 8.8-fold above the $IC_{80}$ (0.46 µg ml$^{-1}$). Remarkably, viral blips were observed with median ITS103.01 levels at 239-fold above the $IC_{80}$ at 12.4 µg ml$^{-1}$, though this ranged between 0.3–103.5 µg ml$^{-1}$. We used the same conservative approach to estimate the challenge leading to the viral blip as that used for sustained viremia above; the median level of ITS103.01 at the causative challenge was 413-fold greater than the $IC_{80}$ (21.5 µg ml$^{-1}$), with a similarly wide range (0.70–146 µg ml$^{-1}$). The earliest viral blip occurred more than 3 weeks after initial mAb infusion; hence, mAb penetration of mucosal tissues was expected[29]. These viral blips suggest that subclinical infections may be possible at bNAb levels above those when breakthrough infection occurs.

## Fully neutralizing mAbs delay viremia

A second study in NHPs was designed to better elucidate which SIV challenge(s) caused viral blips and sustained viremia, such that the exact concentrations of infused neutralizing mAbs at breakthrough could be characterized. Animals ($n = 6$ per group) were infused with low-potency ITS102.03 (which completely neutralizes SIVmac239 virus; Extended Data Fig. 1a) or high-potency ITS103.01 (Fig. 3a). The potencies of these two CD4bs mAbs differ by approximately 130-fold in terms of their in vitro SIVmac239 neutralization $IC_{80}$ values (3.0 µg ml$^{-1}$ versus 0.022 µg ml$^{-1}$). Both antibodies bind near or on the CD4bs. Animals were then intrarectally challenged weekly with distinct individual molecularly tagged (barcoded) but otherwise isogenic and phenotypically equivalent variants of the difficult-to-neutralize tier 3 virus SIVmac239 (ref. 30) until sustained viremia of at least 1,000 copies per millilitre occurred. One of eight uniquely barcoded viruses was used at each challenge, such that sequencing of the eventual viremia would indicate which challenge gave rise to infection. ITS102.03 and ITS103.01 treatment delayed viremia relative to controls (Fig. 3b), with significantly greater times to reach a viral load of more than 1,000 copies per millilitre (each $P = 0.006$) (Fig. 3c). The delay in acute viremia was also greater for ITS103.01- versus ITS102.03-treated animals. When the viral load curves were synchronized, the presence of mAbs largely did not affect the set-point viral load (Fig. 3d). However, both ITS102.03- and ITS103.01-infused animals showed a reduced upslope of viremia onset ($P = 0.015$ and $P = 0.030$, respectively) and a lower peak viral load ($P = 0.0058$ and $P = 0.036$, respectively) relative to control animals (Fig. 3e,f). There was no difference in upslope or peak viral load between the two mAbs.

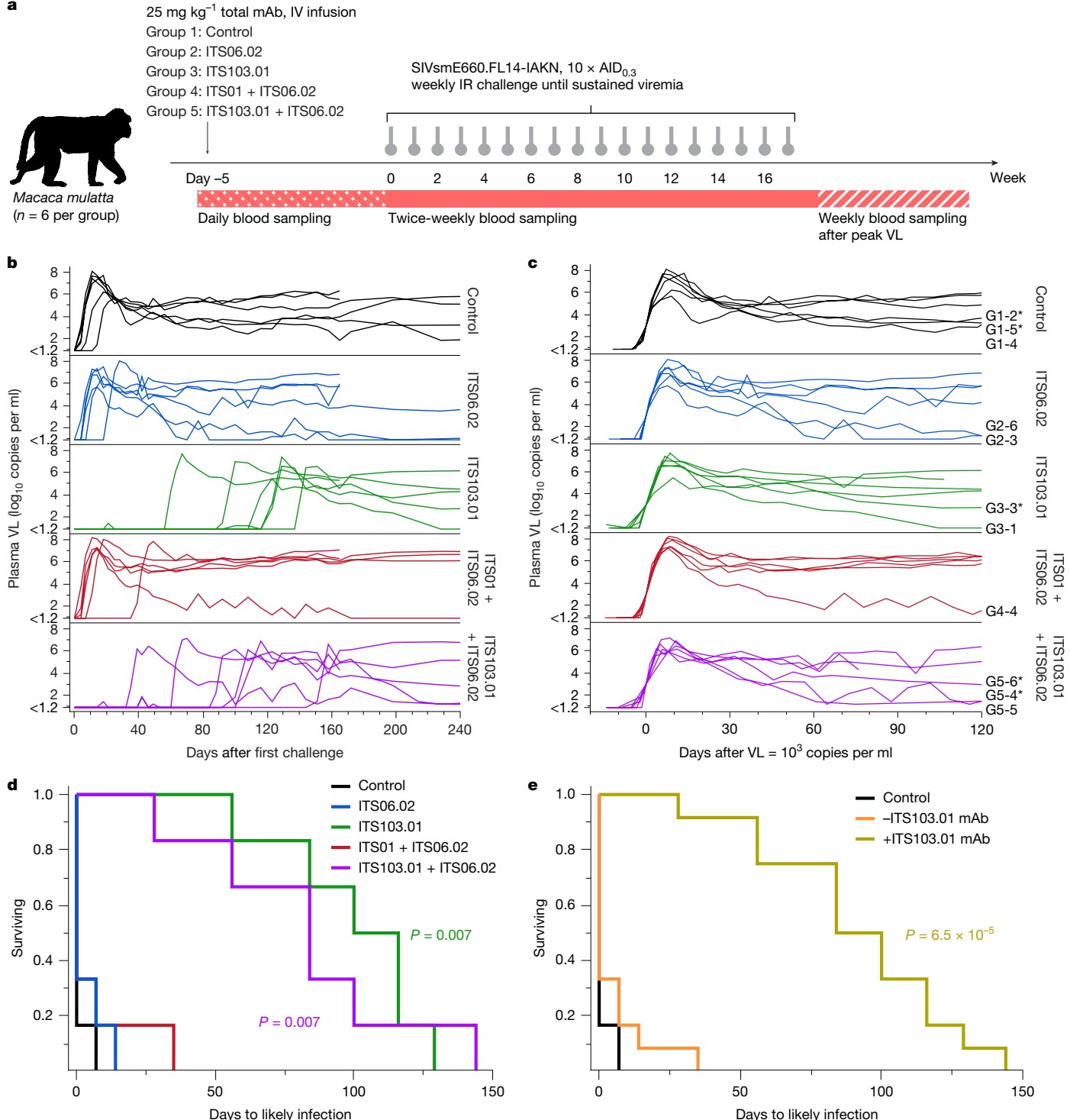

**Fig. 1 | Study design, SIV viral loads and time to infection following fully and partially neutralizing antibody infusion. a**, Infusion of partially and/or fully neutralizing mAbs and schema of the challenge study. **b**, Longitudinal plasma viral load (VL) up to 240 days after the first challenge, stratified according to treatment group. **c**, Longitudinal plasma VL synchronized to when 1,000 copies per millilitre was reached, stratified according to treatment group. Animals with VLs trending lower over time have their IDs indicated next to their curve. IDs with an asterisk indicate animals with known MHC alleles associated with control of virus replication (Supplementary Table 5). **d**, Survival curves of days until likely infection date according to

treatment group. **e**, Survival curves of days until likely infection, with animals from antibody treatment groups clustered according to whether they received treatment with the fully neutralizing ITS103.01 mAb (+ITS103.01 mAb) or lacked ITS103.01 in the delivered mAbs (−ITS103.01 mAb). Group 1 (control) animals are shown separately. In **d** and **e**, survival curves for the control and each mAb treatment group or cluster were compared using a two-tailed Wilcoxon test; significant differences ($P < 0.05$, Bonferroni-corrected for multiple comparisons) are indicated for each mAb treatment (by legend colour) versus control.

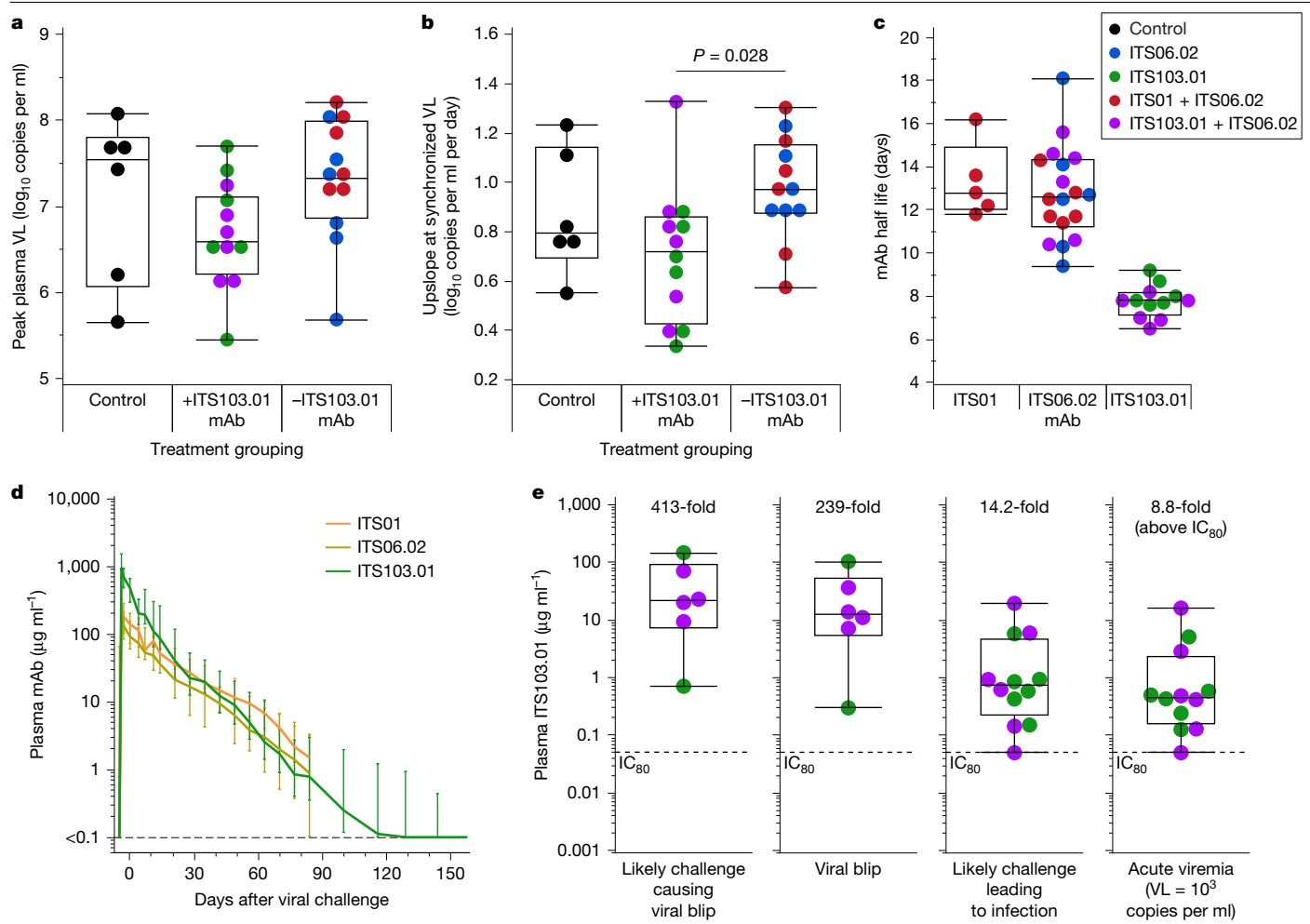

**Fig. 2 | Viral kinetics and plasma antibody levels following fully and partially neutralizing antibody infusion. a,b**, Peak plasma VL (**a**) and rate of viremia increase (upslope; **b**) measured from synchronized VL curves when VL was equal to 1,000 copies per millilitre. Animals in the antibody treatment group were pooled according to whether they received the fully neutralizing ITS103.01 (groups 3 and 5) or the partially neutralizing mAbs (groups 2 and 4). Group 1 (control) animals are shown separately. For each graph, groups were compared by two-tailed one-way analysis of variance, and all pairwise comparisons were assessed using Tukey's honest significant difference (HSD) test; significant (adjusted $P < 0.05$) differences are indicated. **c,d**, Antibody half-life in days of infused mAbs (**c**) and median longitudinal plasma mAb levels (**d**) (error bars indicate range; $n = 6$ animals for ITS01, $n = 18$ for ITS06.02, $n = 12$ for ITS103.01) with respect to the day of first viral challenge. For animal G4-3,

no ITS01 half-life is shown, and the longitudinal plasma mAb levels for this animal were excluded from the median as high endogenous reactivity to the ITS01 anti-idiotypic antibody prevented calculation of the mAb decay curve (Extended Data Fig. 1). **e**, Plasma ITS103.01 levels measured at the time of the likely challenge causing a viral blip, at the time of the viral blip, at the challenge that probably led to infection and at the time of acute viremia (VL equal to 1,000 copies per millilitre). In each graph, the fold increase of the median mAb concentration relative to the ITS103.01 $IC_{80}$ (marked by a dotted line for reference) is indicated above the data points (Extended Data Figs. 1 and 2 and Supplementary Table 1). In **a**–**c** and **e**, data points for individual animals are shown and coloured according to treatment group ($n = 6$ animals per group in total). The overlaid boxes and middle line indicate the interquartile range (IQR) and median, respectively, and the whiskers indicate the range.

## Subclinical infections despite potent mAbs

Barcode sequencing showed that all control animals became infected at the first challenge, with only barcode A detected in their plasma virus (Supplementary Table 2). Half of the ITS102.03-treated animals also became infected at the first challenge, whereas the others were infected after the second challenge (barcode B) (Supplementary Table 3). Therefore, ITS102.03 infusion did not significantly delay virus infection versus controls (Fig. 4b), despite ITS102.03-treated animals showing delayed acute viremia. One ITS102.03-treated animal, B2-5, incurred a secondary infection with virus C, as evidenced by a low abundance (0.02%) of barcode C sequences at day 32 (Supplementary Table 3). ITS103.01-treated animals all became infected with a single dominant barcode, characterized as a barcode with more than 50% abundance at all time points once sustained viremia had occurred (Fig. 4a and Supplementary Table 4). For animals B3-2, B3-3 and B3-5,

the dominantly infecting barcode A was more likely to have come from the second challenge at day 56 rather than from day 0, as sustained viremia did not occur until day 67–71 (11–15 days after the second barcode A challenge). When considering the infection date of dominant viruses, ITS103.01 treatment significantly delayed virus infection versus controls ($P = 0.008$) (Fig. 4b). Similar to the first NHP study using an easier-to-neutralize tier 2 virus, five of six ITS103.01-treated animals showed one or more viral blips. Although several attempts were made to determine the barcodes of the viremia at these blips, most were not recovered. However, barcode F was observed for the viral blip of animal B3-4 at day 35, 1 week following challenge with virus F; this was distinct from the barcode H that dominated the later viremia in this animal, suggesting that a subclinical infection had occurred.

Low-abundance barcodes were also detected for all ITS103.01-treated animals during sustained viremia. Animals B3-1, B3-2, B3-4 and B3-6 showed barcodes that indicated secondary infections (barcodes H, B, I

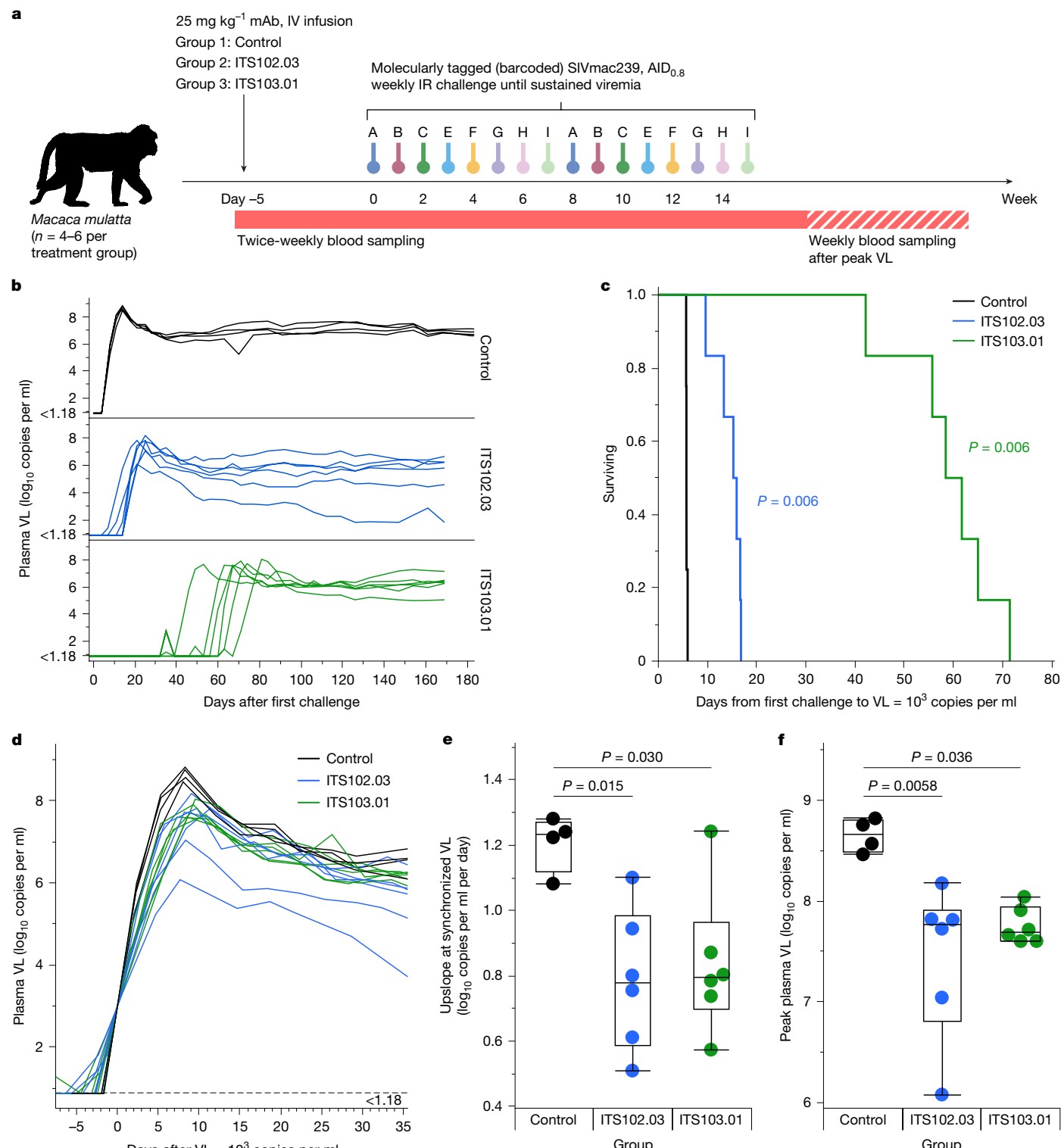

**Fig. 3 | Barcoded SIV challenge study design, SIV VLs and kinetics. a**, Barcoded SIV challenge study schema. **b**, Longitudinal plasma VL up to 180 days after the first challenge, stratified according to treatment group. **c**, Survival curve for days from first challenge to acute viremia (first occurrence of VL equal to 1,000 copies per millilitre) for each treatment group. All pairwise comparisons were determined by two-tailed Wilcoxon testing, and significant differences ($P < 0.05$, Bonferroni-corrected for multiple comparisons) are indicated for each mAb treatment (by legend colour) versus control. **d**, Longitudinal plasma VL synchronized to when a VL of 1,000 copies per millilitre was first reached. Curves for individual animals

are coloured according to treatment group. The limit of detection (15 copies per millilitre) is indicated by the dotted line. **e,f**, Rate of viremia increase (upslope; **e**), measured from synchronized VL curves when VL was equal to 1,000 copies per millilitre, and peak plasma VL (**f**). For each graph, groups were compared by two-tailed one-way analysis of variance, and all pairwise comparisons were assessed using Tukey's HSD test. Significant (adjusted $P < 0.05$) differences are indicated. $n = 4$ (control group) or $n = 6$ (ITS102.03 and ITS103.1-treatment groups) animals. The overlaid boxes and middle line indicate the IQR and median, respectively, and the whiskers indicate the range.

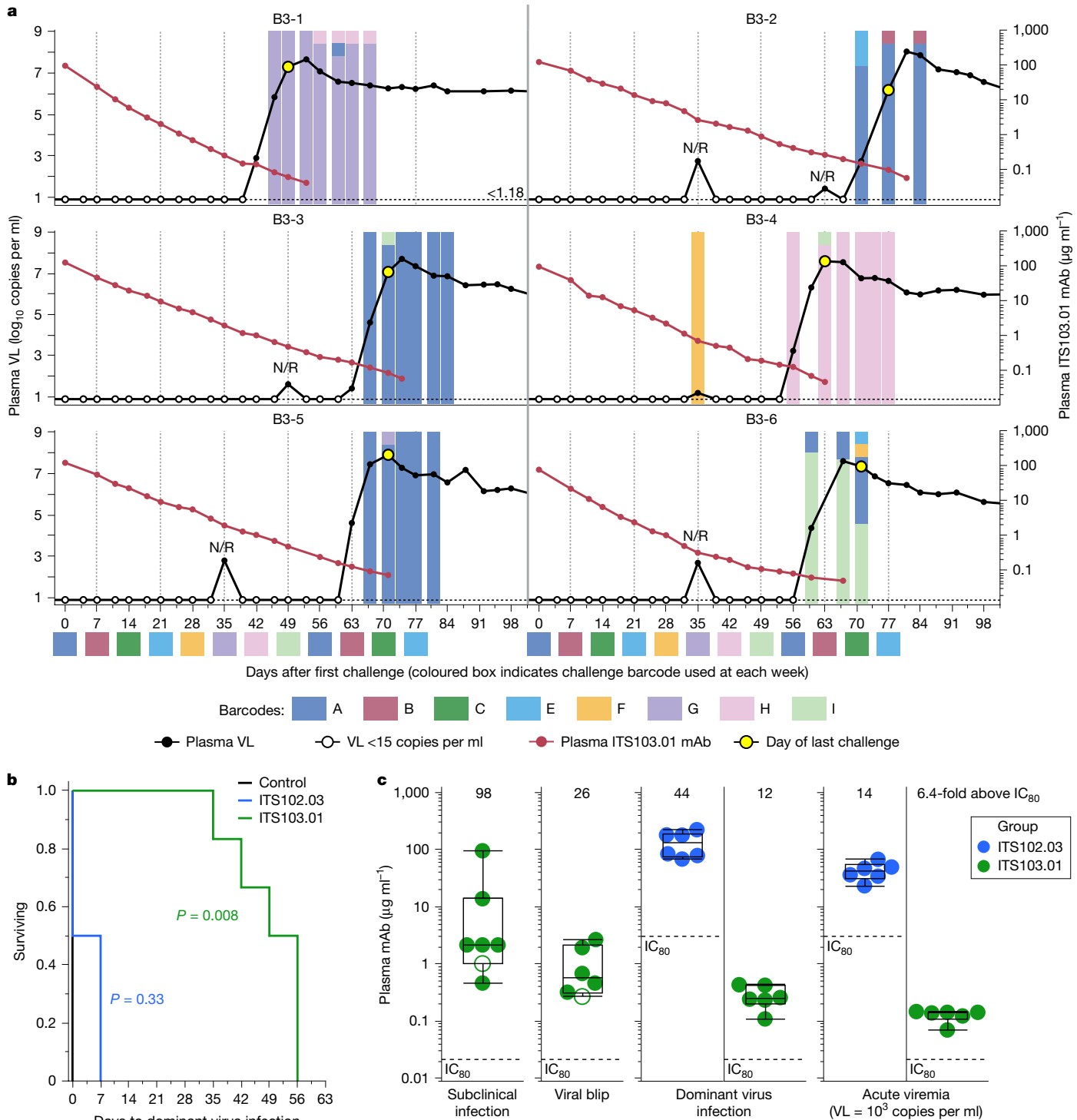

**Fig. 4 | Virus infections and plasma antibody levels with respect to plasma viremia barcode sequencing. a**, Longitudinal plasma VL (black) and plasma ITS103.01 mAb levels (red) up to 98 days after the first challenge are overlaid with each panel for individual animals in group 3. Open black circles represent VL of less than 15 copies per millilitre. The coloured boxes under the horizontal axis numbers designate the virus barcode used at each weekly challenge, with each animal's final challenge indicated (yellow-filled circle). Time points where plasma viremia barcode sequencing was performed are marked by coloured bars aligned to a VL data point. Colours within the bar indicate the specific virus barcodes detected, and the area of each colour is proportional to the relative abundance of the given barcode (except for abundances <1%, which are shown as squares within their bar). N/R over a data point indicates that no barcode sequence was recovered (Supplementary Table 4). **b**, Survival curve of days to

dominant virus infection for each treatment group. All pairwise comparisons were determined by two-tailed Wilcoxon testing, and significant differences ($P < 0.05$, Bonferroni-corrected) are indicated for each mAb treatment (by legend colour) versus control (Supplementary Tables 1–3). **c**, Plasma mAb levels measured at a subclinical infection, at viral blip, at the dominant virus infection and at acute viremia (VL = 1,000 copies per millilitre), for animals in group 2 or 3 where appropriate ($n = 6$ animals per group total). In each graph, the fold increase in the median mAb concentration relative to the ITS103.01 or ITS102.03 $IC_{80}$ (marked by a dotted line for reference) is indicated above the data points. Individual animal data points are coloured by treatment group. Open circles represent secondary subclinical infections or viral blips for a given animal. The overlaid boxes and middle line indicate the IQR and median, respectively, and the whiskers indicate the range (Extended Data Fig. 3).

and A) occurring the week after the challenge leading to their dominant infection (barcodes G, A, H and I, respectively). Given the lag between collecting samples for viral load measurement and determining the result, animals were typically challenged a further 2–3 times after the challenge causing the dominant infection. Evidence of subclinical infections occurring before the dominant virus challenge was seen in animals B3-3 and B3-5, with low-copy barcodes (I and G, respectively) at or close to peak viremia; these infections were probably caused by challenges 1 and 3 weeks, respectively, before the dominant infection with virus A at day 56. Animal B3-2 showed an even earlier subclinical infection, with barcode E being detected at day 70 despite that virus having been used for challenge at day 21 (5 weeks before the dominant virus A infection occurred). Similarly, barcode E was observed in animal B3-6 on day 70. The same B3-6 time point also showed a low abundance of barcode F, which was used for challenge at day 28, with a blip (but no sequence confirmation) following this challenge at day 35. In the most extreme case, barcode A was observed in animal B3-1 after peak viral load had occurred (owing to the dominant barcode G infection), but the last virus this animal was challenged with was virus I; this demonstrates that the low-copy plasma barcode A was caused by a subclinical infection from the first challenge given more than 8 weeks earlier, at the highest levels of bNAb. Overall, each ITS103.01-treated animal demonstrated some evidence of subclinical infection before its dominant viral challenge. The lack of complete concordance between detection of transient plasma viral load blips and recovery of viral sequence was probably due, at least in part, to limitations of available samples and sensitivity of analyses.

ITS103.01 levels at the time of subclinical infection were a median of 98-fold greater than the $IC_{80}$ (2.16 μg ml$^{-1}$) but had a wide range of 0.47–96 μg ml$^{-1}$, with most of the infections occurring below 10 μg ml$^{-1}$ (Fig. 4c and Extended Data Fig. 3). Median ITS103.01 levels were 26-fold greater than the $IC_{80}$ while viral blips were occurring. Plasma ITS103.01 had a median level of 0.252 μg ml$^{-1}$ at the time of dominant virus infection, which decreased to 0.142 μg ml$^{-1}$ at acute viremia (these levels were 12- and 6.4-fold greater than the $IC_{80}$, respectively). Although ITS102.03-treated animals did not show a significant delay of infection, their mAb levels were elevated to comparable levels: 44-fold and 14-fold greater than the $IC_{80}$ at dominant infection and at acute viremia, respectively. Together, these data show that plasma neutralizing activity measured in vitro is an excellent correlate of bNAb efficacy, for either levels allowing blips or breakthrough infection.

## Limited immune response from mAbs or blips

Endogenous binding antibody responses against SIVmac239 Env were assessed from week 0 following the dominant virus infection. Antibody titres did not increase above background in any group until more than 2 weeks after infection (Extended Data Fig. 4a). For ITS103.01-treated animals, this suggested that the viral blips were insufficient to induce antiviral IgG before sustained viremia. The median binding responses across all three groups were not substantially different up to 16 weeks postinfection (Extended Data Fig. 4b). The SIVmac239 Gag- and Env-specific T cell response from 2–16 weeks after dominant infection was also assessed; it did not show increased magnitude or earlier occurrence in mAb-treated animals (Extended Data Fig. 5a,b). Responses were Th1 biased, with fewer than 0.5% antigen-specific memory CD4$^+$ or CD8$^+$ T cells for most animals by week 16. Cellular responses 2 weeks after each viral blip were also assayed (Extended Data Fig. 5c,d). Notably, animal B3-2, which had two viral blips (one early at day 35 and the other close to the start of sustained viremia at day 63), showed Env-specific CD8$^+$ T cell responses at both times (0.1% and 0.2%, respectively). Whereas the second time point assayed fell within the onset of sustained viremia, and thus the response was most likely to have been driven by the increasing amount of viral antigen, the first viral blip alone seemed to have been sufficient to elicit the small T cell response observed.

## Limited escape mutations due to mAbs

The occurrence of Env mutations in regions containing the mAb epitopes[28] was assessed at peak viral load for all animals (or at further time points surrounding the peak for some animals). Minimal mutations were observed in control animals, with only around 0.1% of sequences showing alterations in the 365 potential N-glycan sites (PNGSs) (Extended Data Fig. 6a; HXB2 numbering used here and subsequently). Similar proportions of these mutations at the same site were observed in several mAb-treated animals. B3-6 showed mutations in V4 7 days before peak viral load, but these were absent 4 days after the peak. ITS102.03-treated B2-2 had a majority of reads lacking the 280 PNGSs at peak viral load. The ITS103.01-treated animal, B3-2, had the same mutation although less abundantly. In addition, 63% of the reads for B3-2 at site 465 showed an absent PNGS. In both cases, mutations of B3-2 were minimally evident 4 days before peak viral load (3% of reads) but predominant at the peak and 3 days after (65% of reads). Testing of the neutralization sensitivity of these two PNGS mutations confirmed that N280D prevented both ITS102.03 and ITS103.01 neutralization, whereas N465S (observed only in a ITS103.01-treated animal) was only effective at preventing ITS103.01 neutralization (Extended Data Fig. 6c).

Given that animal B3-2 was subclinically infected with virus E before dominant infection with virus A and secondary virus B infection, single-genome analysis was used to determine whether these PNGS mutations were linked to each other and/or to a specific barcode. At peak viral load, 25 of 30 isolates contained N465S/D, and two showed T297A/I (which removed the 280 PNGS), whereas the remaining three were wild type (Extended Data Fig. 6b). All the single-genome analysis isolates bore barcode A. At 12 weeks after peak viral load, all the genomes sequenced were wild type at the 280 and 465 PNGSs, suggesting that these glycan mutations were transient as ITS103.01 levels decreased following breakthrough infection.

## Discussion

Administration of bNAbs could be a valuable tool for prevention of HIV acquisition. Here, we sought to characterize the plasma concentrations of rhesus bNAbs required for protection from SIV infection in rhesus macaques. In an initial study using partially neutralizing (ITS01, ITS06.02) and fully neutralizing (ITS103.01) mAbs, delayed viremia relative to infection was only achieved when animals received the potent bNAb ITS103.01. Characterization of ITS103.01 levels in animals receiving this mAb (groups 3 and 5) with respect to virological events showed median antibody levels of 14.2-fold greater than the in vitro $IC_{80}$ at the time of the challenge likely to have led to infection. However, the range of individual values spanned more than two orders of magnitude, with mAb levels of one-fold and more than 200-fold the $IC_{80}$ observed (Fig. 2e). These results indicate that estimating infection date on the basis of the occurrence of acute viremia may limit determination of accurate protective bNAb levels, given that the time between virus exposure and detectable viremia in plasma varied considerably between animals.

Our second study using uniquely barcoded virus challenges each week facilitated accurate determination of the virus exposure leading to detected plasma viremia. The benefit of this approach was highlighted by comparisons of the delays in virus infection following administration of the low-potency ITS102.03 and high-potency ITS103.01 bNAbs: comparison of the time taken for viral load to reach 1,000 copies per millilitre showed significant delays for both bNAbs (Fig. 3c), whereas using viral barcode sequencing to define the challenge virus leading to sustained viremia demonstrated that only ITS103.01-treated animals had significantly delayed infection (Fig. 4b). The barcoded viruses also yielded evidence of subclinical infections occurring in animals treated with ITS103.01. In five of six animals, low-abundance barcodes were detected during or soon after acute viremia; this suggested that virus

challenges before the dominant virus infection had caused infection but did not yield detectable viremia while bNAb levels were sufficiently high. In addition, a barcode was recovered for a viral blip in the sixth animal (animal B3-4, Fig. 4a) and was consistent with the blip having originated from the virus challenge delivered 1 week earlier. A limitation of this study was its failure to confirm the virus barcode causing viral blips in the other animals, despite repeated sequencing attempts. Given that twice-weekly bleeds were collected, only a relatively small volume of plasma was available for viral load testing and, if positive, subsequent sequencing. Future studies of viral blips may benefit from optimized workflows for viral load and barcode determination. The time between challenge and plasma barcode detection ranged from 1 to 8 weeks; this points to the plasma virus consisting of progeny virions from infected host cells rather than virions from the inoculum, given that the mean lifespan of plasma virions is less than a day[31]. Persistence of infected cells while neutralizing mAb levels are protective is consistent with a report of cell-associated SHIV challenge following PGT121 mAb infusion in NHPs, in which one animal was observed to have a 7-week delay of viremia[32]. In the present study, the median plasma ITS103.01 bNAb concentration at the time of initiation of these subclinical infections was 98-fold greater than the in vitro $IC_{80}$, with two of seven of the barcode-evidenced events occurring at mAb levels greater than 10 µg ml$^{-1}$ml (approximately 450-fold greater than the $IC_{80}$) (Fig. 4c). This value was in agreement with analyses of the AMP trials, which reported 90% protective efficacy when VRC01 levels were approximately 200-fold greater than the $IC_{80}$ (ref. 23).

Many of the animals treated with ITS103.01 bNAb in both immuno-prophylaxis studies exhibited viral blips between their first challenge and the onset of sustained viremia, specifically, half of the ITS103.01-treated animals in the first study and five of six animals in the barcoded virus study. The ITS103.01 bNAb was the only mAb to significantly delay virus infection; therefore, the lack of observed viral blips with the other mAbs used is likely to have been the product of an insufficient protection period between the first challenge and breakthrough virus infection. So far, viral blips such as these have not been described in humans undergoing immunological or small-molecule prophylactic treatment against HIV infection. It is, therefore, unclear whether this phenomenon is unique to NHPs and SIV or a general behaviour of HIV and/or SIV. However, the plasma sampling schedule used for the animal experiments in this study (twice-weekly blood draws) involved far more frequent sampling than typically occurs with human cohorts for practical reasons. Given that most of the viral blips observed were present only in one of two samples in each week, it would be difficult to observe viral blips in human prevention studies owing to probable lower sampling frequency. In addition, the threshold sensitivity of monitoring assays used for HIV testing in humans is generally around 40 copies per millilitre[33], whereas the assays used here provide a threshold sensitivity of 15 copies per millilitre[34]; this may also have contributed to greater detection of viral blips here. Our data indicate that maintenance of mAb levels exceeding the breakthrough concentration level may be crucial to the success of any antibody-mediated treatment. Such levels are sufficient to suppress viremia (even following a detectable plasma viral RNA blip) and keep replication below that required to generate escape mutants. It is unknown how long this suppression would have to be maintained to eliminate the virus, if this were possible. Animal B3-1 showed viremia with a challenge virus to which it had been exposed more than 8 weeks earlier.

Many in the prevention field are currently working towards assessing whether prophylactic administration of two or more bNAbs can effectively prevent HIV acquisition, with several clinical trials assessing mAb combinations[16,21,35], and these bNAb combinations will certainly be beneficial in preventing infections from circulating isolates bearing resistance mutations to a single bNAb or bNAb class. However, the data in this study indicate that combination bNAbs may be insufficient to prevent subclinical infections. The observed plasma viral blips represented acute viral replication occurring in the animals (after challenge at

a mucosal site), which was then suppressed by the high concentrations of ITS103.01 bNAb present. Therefore, this virus was sensitive to the bNAb and did not show the sustained viremia that would be likely to result from development of mAb escape mutations. Indeed, sequencing of the *env* gene from plasma virus at around the time of peak viral load showed that only one of six animals had virus that developed ITS103.01 escape mutations, and that this mutation was transient—that is, it was seen at peak viral load, but not 4 days earlier, and it was absent 12 weeks later. Whether or not these subclinical infections are capable (once bNAb levels decline) of causing typical acute viremia onset was not assessed in this study; therefore, further investigation is needed to address this question.

The endogenous immune response of animals in the barcoded virus study was assessed to determine whether bNAb treatment exerted a 'vaccinal effect', in which endogenous cellular and humoral responses are enhanced by the presence of an antiviral antibody. There was no increase in the magnitude or rate of onset of T cell responses against Gag or Env, nor was the Env-specific antibody response enhanced by the presence of ITS103.01 or ITS102.03. This is in contrast to other studies in SHIV-infected NHPs given HIV bNAbs therapeutically[36–39] and may indicate that induction of a vaccinal response probably requires an existing memory response or substantial viral antigen to be present when mAb is first injected. In addition, the viral Env antigens differ between those used in these therapeutic SHIV bNAb studies (that is, HIV-derived Env) and the SIVmac239 Env expressed by viruses used here.

Long-lasting, broad and potent neutralizing antibodies will be required for HIV-1 prevention strategies to successfully overcome the diversity in circulating HIV-1 isolates in the human population. Initial human clinical trials have shown that bNAbs can effectively prevent infections from susceptible virus, so better defining the levels of bNAb required for protection in model systems will benefit future studies and clinical work. Here, using an NHP and SIV model, we demonstrated that fully neutralizing bNAbs can efficiently delay infection, but viral blips could be observed throughout this protection window. Further interrogation of these blips using barcoded challenge viruses provided evidence of subclinical infections occurring while bNAb levels were sufficiently high to typically prevent virus replication. It is important for these viral blips to be further investigated with respect to whether they occur with HIV-1 in humans even in the absence of T cell responses. If so, this would be likely to complicate the effectiveness and interpretation of bNAb immunoprophylaxis studies in humans.

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

## Methods

### Cell lines and cell culture

HEK 293T cells (ATCC) and TZM-bl cells (NIH HIV Reagent Program) were maintained in Dulbecco's modified Eagle medium (Thermo Fisher Scientific) supplemented with 10% (v/v) heat-inactivated fetal bovine serum (Gibco) and 1% penicillin–streptomycin at 37 °C and 5% $CO_2$. Expi293F cells were maintained in Expi293 Expression Medium (Thermo Fisher Scientific) in Erlenmeyer flasks at 37 °C, 5% $CO_2$, 125 rpm. Cell lines were authenticated by the organizations from which they were obtained: 293T cells by STR profiling, Expi293F cells by STR profiling and assessment of cell morphology and growth kinetics, and TZM-bl cells by assessment of cell morphology and functional testing in neutralization assays. Cell lines were not routinely tested for mycoplasma contamination.

### Antibody expression and purification

The ITS01, ITS06.02, ITS102.03 and ITS103.01 (both LS and non-LS versions for the latter two) mAbs were produced by cotransfection of plasmids encoding heavy and light chains for a given antibody at a 1:1 mass ratio into Expi293F cells (Thermo Fisher Scientific) using an Expi-Fectamine 293 Transfection Kit (Thermo Fisher Scientific) according to the manufacturer's instructions. Cells were cultured at 37 °C for 6 days following transfection and then moved to 32 °C for the final day 7 of culture. Culture supernatant was clarified by low-speed centrifugation and 0.22-μm filtration. Antibodies were purified using rProtein A Fast Flow resin (Cytiva), buffer exchanged into phosphate-buffered saline (PBS) and sterile filtered. Antibody preparations were stored at 4 °C or frozen at −80 °C.

### Virus stock production

SIVsmE660.FL14-IAKN virus (a molecular tier 2 clone of SIVsmE660) was initially prepared in HEK 293T cells transfected with SIVsmE660-encoded plasmid pUC19-E660-FL14 using FuGene6 (Promega). Two days posttransfection, the cell supernatant was collected (transfection stock). Rhesus macaque peripheral blood mononuclear cells (PBMCs) were isolated from whole blood using SepMate tubes (StemCell Technologies) with Ficoll (Cytiva) density gradient centrifugation. CD4$^+$ T cells were enriched by positive selection (CD4 MicroBeads, NHP, Miltenyi Biotec). Rhesus CD4$^+$ T cells were activated by incubation with α-CD2/α-CD3/α-CD28 microbeads (T Cell Activation/Expansion Kit, NHP, Miltenyi Biotec) in 30 U ml$^{-1}$ IL-2 in RPMI medium containing 15% (v/v) heat-inactivated fetal bovine serum and 1% penicillin–streptomycin (RF15-IL2) for 4 days. The rhesus macaque PBMCs were separately isolated and activated from 3–5 animals. The activated CD4$^+$ T cells were harvested, pooled and infected with the transfection stock at a multiplicity of infection of 0.1–0.01. Cells were incubated at 37 °C and 5% $CO_2$ for 4 h with gentle mixing every 30 min and washed with R15-IL2 three times. Then, the infected cells were transferred to tissue-culture-treated flasks and cultured for 15 days. During the 15 days, 200 μl of culture supernatant was collected and frozen every 3 days beginning at day 0 for p27 ELISA (Advanced Bioscience Laboratories). About 75% of culture supernatant was collected and frozen at days 6, 9, 12 and 15, and the same volume of warm R15-IL2 was added to the flask. Supernatants with relatively high p27 values were thawed, aliquoted and stored at −80 °C (infection stock). Individual infectious stocks of the molecularly tagged (barcoded) SIVmac239 viruses A to I were generated as described previously[30].

### Animal studies

All experiments were carried out in compliance with National Institutes of Health regulations and with approval from the Animal Care and Use Committee of the Vaccine Research Center and Bioqual, Inc., where NHPs were housed for the duration of the studies. Animals were housed and cared for in accordance with local, state, federal and institutional policies in facilities accredited by AAALAC International under standards established in the Animal Welfare Act and the Guide for the Care and Use of Laboratory Animals. In accordance with the institutional policies of both institutions, all compatible NHPs were always pair-housed; single housing is only permissible when scientifically justified or for veterinary medical reasons, and for the shortest duration possible. NHPs were housed in appropriately sized caging according to the *Guide for the Care and Use of Laboratory Animals*[40] and provided with a variety of enrichment toys, treats, fresh produce and foraging devices. Water was offered ad libitum, and animals were fed primate biscuits (Monkey Diet, 5038, Lab Diet) twice daily. As standard practice, animal holding rooms were maintained on a 12-h light–dark cycle, with a room temperature of 16–21 °C and relative humidity of 30–70%. The investigators were not blinded to the animal treatment allocations.

In the partially and fully neutralizing mAb infusion and challenge study, 30 Indian-origin rhesus macaques (*Macaca mulatta*), aged 2–5 years, were stratified by sex, age, weight and *TRIM5* genotype into five groups of six animals (Supplementary Table 5). Distribution of *TRIM5* genotypes was required as SIVsmE660 acquisition is affected by certain alleles[41,42]. Sample size was chosen to yield at least 85% power to detect a difference in time to infection (this also applied to the barcoded virus study described below). Animals were intravenously infused with ITS06.02 (group 2), ITS103.01 (group 3), ITS01 and ITS06.02 (group 4), or ITS103.01 and ITS06.02 (group 5). All animals received a total of 25 mg kg$^{-1}$ of mAb. For groups 4 and 5, the mAbs were given at a mass ratio of 1:1. The mAbs used all contained the LS mutation[43]. Control animals (group 1) were untreated. Five days after infusion, animals were intrarectally challenged with a limiting dose 10-fold over the animal infectious dose leading to infection of 30% of control animals ($10 \times AID_{0.3}$) of SIVsmE660.FL14-IAKN infection stock. For each animal, these challenges were repeated weekly for a maximum of 17 weeks until sustained viremia occurred. Blood samples were collected throughout the study.

The barcoded virus challenge study used 16 Indian-origin rhesus macaques (*M. mulatta*), aged 2 years, which were stratified by sex, age and weight into one group of four animals (control, group 1) and two groups of six animals (mAb treatment, groups 2 and 3) (Supplementary Table 6). mAb-treatment animals were intravenously infused with ITS102.03 (group 2) or ITS103.01 (group 3) at 25 mg kg$^{-1}$. To decrease plasma half-life to allow faster decline of mAb concentrations, mAbs that did not contain the LS mutation were used here. Control animals were not infused with mAb. Five days following mAb infusion, all groups began receiving weekly intrarectal challenges sequentially using one of eight uniquely molecularly tagged (barcoded) SIVmac239 viruses, in the order A, B, C, E, F, G, H, I. The challenge dose was equal to a dose that infects 80% of control animals ($AID_{0.8}$), and challenges (up to a total of 16 by repeating the same barcodes in a second series) were continued until sustained viremia occurred in each animal. Blood samples were collected throughout the study.

### Viral load measurement

Plasma SIV RNA measurements were performed essentially as described previously[34]. The upslope at synchronized viral load was calculated as the gradient between the plasma measurements immediately before and after viral load first passed through 1,000 copies per millilitre. The set-point viral load in each animal was calculated as the geometric mean of viral load measurements from 45 to 120 days after the viral load first reaching 1,000 copies per millilitre of plasma. Groups were statistically compared using Kruskal–Wallis and Dunn's multiple comparisons test.

### SIVmac239 barcode identification

Plasma virus barcodes were identified using either single-genome amplification as described previously[30] or MiSeq sequencing as described elsewhere[44]. For sequence analysis, we used the Frederick National Laboratory for Cancer Research Barcode Analysis Tool v.2022

## Plasma mAb measurement

Concentrations of infused mAbs in plasma samples were determined by enzyme-linked immunosorbent assays (ELISAs) using anti-idiotypic antibodies. MaxiSorp ELISA plates (96-well, Thermo Fisher Scientific) were coated with 100 µl per well of 2 µg ml[-1] anti-idiotypic antibody (17B4-IgG1, 8A4-IgG1, ITS103-id1 or ITS102-id1, as appropriate for the mAb of interest)[27,28] in PBS (pH 7.4) and incubated overnight at 4 °C. Plates were blocked with 300 µl per well PBS containing 5% (w/v) skim milk powder at 37 °C for 1 h. All subsequent steps involved incubation at the same temperature and for the same time, with 100 µl loaded per well. For ITS01 and ITS06.02 quantitation assays, plasma samples were heat inactivated at 56 °C for 1 h before being diluted into PBS containing 5% (w/v) skim milk powder and 0.05% (v/v) Tween 20 (diluent buffer). For ITS102.03 and ITS103.01 quantitation assays, plasma samples were not heat inactivated before dilution into diluent buffer. Serial dilutions of the appropriate purified ITS mAb into diluent buffer were used as standards in the assay. After blocking, wells were washed five times with PBS containing 0.1% Tween 20 (and similarly washed after each subsequent step). Dilutions of plasma or mAb were then added to the plate. Bound antibody was detected using mouse anti-monkey IgG-HRP (Southern Biotech) at 1:5,000 in diluent buffer. Reactions were developed using 100 µl per well SureBlue TMB 1-Component Microwell Peroxidase Substrate, with incubation for 10 min at room temperature before addition of 100 µl per well 0.5 M sulfuric acid. Absorbance at 450 nm was measured using a SpectraMax 384 Plus Absorbance Plate Reader and analysed using GraphPad Prism v.10.0.3 (GraphPad Software). Concentrations of infused antibodies were interpolated from the standard curve using a sigmoidal four-parameter curve. Interpolated values from multiple dilutions of each plasma sample were averaged to give the final concentration.

## Endogenous anti-SIV Env antibody assays

Endogenous antibody titres against Env were measured by ELISA using an anti-kappa specific antibody such that no infused mAb in the plasma sample would be detected (both ITS102.03 and ITS103.01 have lambda light chains). High-binding, half-area microplates (96-well, Corning Inc.) were coated with 50 µl per well of 1 µg ml[-1] SIVmac239-foldon trimer[26] in PBS, followed by incubation overnight at 4 °C. Plates were blocked with 150 µl per well PBS containing 5% (w/v) skim milk powder at 37 °C for 1 h. All subsequent steps involved incubation at the same temperature and for the same time, with 50 µl loaded per well. After blocking, wells were washed five times with PBS containing 0.1% Tween 20 (and similarly washed after each subsequent step). Plasma samples from weeks 0 to 16 after the dominant virus infection were assayed, as well as pre-immune plasma (from the day of first virus challenge). Heat-inactivated plasma samples were 1:5 serially diluted into diluent buffer starting from 1:100 (eight points total) before being added to the plate. Bound antibody was detected using goat anti-human Igκ-HRP (Millipore Sigma) at 1:2,500 in diluent buffer. Reactions were developed using 50 µl per well SureBlue TMB 1-Component Microwell Peroxidase Substrate (SeraCare), with incubation for 10 min at room temperature before addition of 50 µl per well 0.5 M sulfuric acid. Absorbance at 450 nm was measured using a SpectraMax 384 Plus Absorbance Plate Reader (Molecular Devices) and analysed using GraphPad Prism. Area under the curve values were calculated for serially diluted plasma samples. These values were normalized by subtracting each animal's pre-immune area under the curve value.

## Neutralization assays

SIV Env pseudotyped viruses were produced by cotransfection of 293T cells with plasmid DNA encoding SIV gp160 along with a luciferase reporter plasmid in the SG3 backbone containing HIV structural genes as described previously[26]. SIV gp160-encoding plasmids for the clone SIVmac239.cs.23 (ref. 45) were provided by D. Montefiori. Plasmids encoding SIV gp160 sequences with amino acid substitutions were synthesized by Genscript. Virus neutralization was measured using infection of TZM-bl target cells by pseudotyped virus as previously described[46]. Titres were calculated as either the $IC_{50}$ or $IC_{80}$ for mAbs that caused a 50% or 80% reduction of infection compared with virus alone using a logistic five-point model (JMP 15).

## Intracellular cytokine staining

Cryopreserved PBMCs were thawed and rested overnight in a 37 °C, 5% $CO_2$ incubator. The next morning, cells were stimulated with SIVmac239 Gag and Env peptide pools (15-mers overlapping by 11 amino acids; provided by the NIH HIV Reagent Program) at a final concentration of 2 µg ml[-1] in the presence of 3 mM monensin for 6 h. Peptide pools were reconstituted in 100% dimethyl sulfoxide. Negative controls received equal concentrations of dimethyl sulfoxide instead of peptides (final concentration of 0.5%). Intracellular cytokine staining was performed as described previously[47]. The following mAbs were used: CD3 APC-Cy7 (clone SP34.2, BD Biosciences), CD4 PE-Cy5.5 (clone SK3, Thermo Fisher), CD8 BV570 (clone RPA-T8, BioLegend), CD45RA PE-Cy5 (clone 5H9, BD Biosciences), CCR7 BV650 (clone G043H7, BioLegend), CXCR5 PE (clone MU5UBEE, Thermo Fisher), CXCR3 BV711 (clone 1C6/CXCR3, BD Biosciences), PD-1 BUV737 (clone EH12.1, BD Biosciences), ICOS Pe-Cy7 (clone C398.4 A, BioLegend), CD69 ECD (cloneTP1.55.3, Beckman Coulter), IFN-g Ax700 (clone B27, BioLegend), IL-2 BV750 (clone MQ1-17H12, BD Biosciences), IL-4 BB700 (clone MP4-25D2, BD Biosciences), TNF-FITC (clone Mab11, BD Biosciences), IL-13 BV421 (clone JES10-5A2, BD Biosciences), IL-17 BV605 (clone BL168, BioLegend), IL-21 Ax647 (clone 3A3-N2.1, BD Biosciences) and CD154 BV785 (clone 24-31, BioLegend). LIVE/DEAD Fixable Aqua Dead Cell Stain Kit (Thermo Fisher Scientific) was used to exclude dead cells. All antibodies were titrated to determine the optimal concentration as reported previously[47]. Samples were acquired on a BD FACSymphony flow cytometer and analysed using FlowJo v.10.8.2 (see Extended Data Fig. 7 for the gating strategy).

## Single-genome amplification sequencing

Viral RNA was isolated from plasma using a QIAamp Viral RNA kit (Qiagen). Extracted RNA was reverse transcribed into complementary DNA (cDNA) using SuperScript III (Qiagen) and viral-specific primer nFL-R1 (5′-CACTAGCTTACTTCTAAAATGGCAGC-3′). The cDNA was then diluted to a single-genome template before PCR with nFL-R1 and nFL-F1 (5′-GATTGGCGCCYGAACAGGGACTTG-3′) primers. Second-round PCR was performed with nFL-R2 (5′-TACTTCTAAAATGGCAGCTTTATTGAA-3′) and nFL-F2 (5′-GTGAAGGCAGTAAGGGCGGCAGG-3′) primers. Both PCR reactions were amplified with Platinum SuperFi DNA polymerase (Thermo Fisher Scientific), and correct-sized amplicons were directly sequenced using BigDye Terminator Sanger sequencing (Thermo Fisher Scientific) with multiple virus-specific primers. Sequences were aligned using Geneious Prime v.2023.0.

## Env MiSeq

An Illumina-based sequencing approach was implemented on a MiSeq instrument to query the a4b7 binding site and V2V3 regions of the viral genome, as previously described[48]. We generated cDNA as described above with the virus-specific primer a4b7.cDNA (5′-TTCTGCCACCTCTGCACTCATGG-3′). MiSeq PCR was performed using a4b7.P7 (5′-TAGAACTTATATTTACTGGCATGG-3′) and a4b7.P5 (5′-GCCACCTCTGCACTCATGG-3′) primers. cDNA for the V2V3 region was generated using the V2.V3.cDNA primer (5′-CCTATCATTGATTGGTTGTGAGTG-3′), and PCR was conducted using V2.V3.P7 (5′-CAGGATAATTGCACAGGCTTGG-3′) and V2.V3.P5 (5′-GCCACCTCTGCACTCATGG-3′).

## Statistical analysis

Data were analysed and plotted using JMP 15 and GraphPad Prism v.10.0.3 with statistical tests as detailed in figure legends or text. $P < 0.05$ was considered to indicate statistical significance (adjusted for multiple comparisons as appropriate).

## Reporting summary

Further information on research design is available in the Nature Portfolio Reporting Summary linked to this article.

## Data availability

The single-genome sequence data have been deposited in GenBank with accession codes PP458374–PP458484 (http://www.ncbi.nlm.nih.gov/nuccore/). The MiSeq data have been deposited with links to BioProject accession number PRJNA1086015 in the NCBI BioProject database (https://www.ncbi.nlm.nih.gov/bioproject/). The other data that support the findings of this study are available from the corresponding author upon reasonable request. Source data are provided with this paper.

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

**Acknowledgements** We thank the NHP care staff at the Vaccine Research Center, NIH and Bioqual, Inc. for their expert animal care and Annie Austin for providing technical assistance with preparation of challenge virus stocks. We also thank the Nonhuman Primate Immunogenicity Core of the Vaccine Research Center, NIH for specimen processing, the Viral Evolution Core and the Quantitative Molecular Diagnostics Core of the AIDS and Cancer Virus Program, Frederick National Laboratory for Cancer Research for viral sequencing and viral load testing. The TZM-bl cell line was obtained through the NIH HIV Reagent Program, BEI Resources Repository, NIAID, NIH. This project was funded in part with federal funds from the National Cancer Institute, National Institutes of Health, under Contract No. 75N91019D00024/HHSN261201500003I. The content of this publication does not necessarily reflect the views or policies of the Department of Health and Human Services, nor does mention of trade names, commercial products, or organizations imply endorsement by the US Government.

**Author contributions** C.A.G., S.K., R.A.K. and M.R. conceived experiments. S.K., N.I., C.M.F. and B.F.K. generated virus challenge stocks. C.A.G., H.A.D.K. and R.D.M. produced antibodies. J.T. coordinated the animal studies. J.D.L. analysed viral load data. C.A.G. and H.A.D.K. evaluated infused antibody levels. H.A.D.K., A.H. and R.D.M. performed and analysed neutralization assays. C.M.F. and B.F.K. sequenced virus barcodes and Env regions. C.A.G., D.R.F. and K.E.F. analysed endogenous immune responses. C.A.G., R.A.K. and M.R. wrote the original manuscript draft. All authors reviewed and edited the manuscript.

**Competing interests** The authors declare no competing interests.

## Additional information

**Correspondence and requests for materials** should be addressed to Mario Roederer.

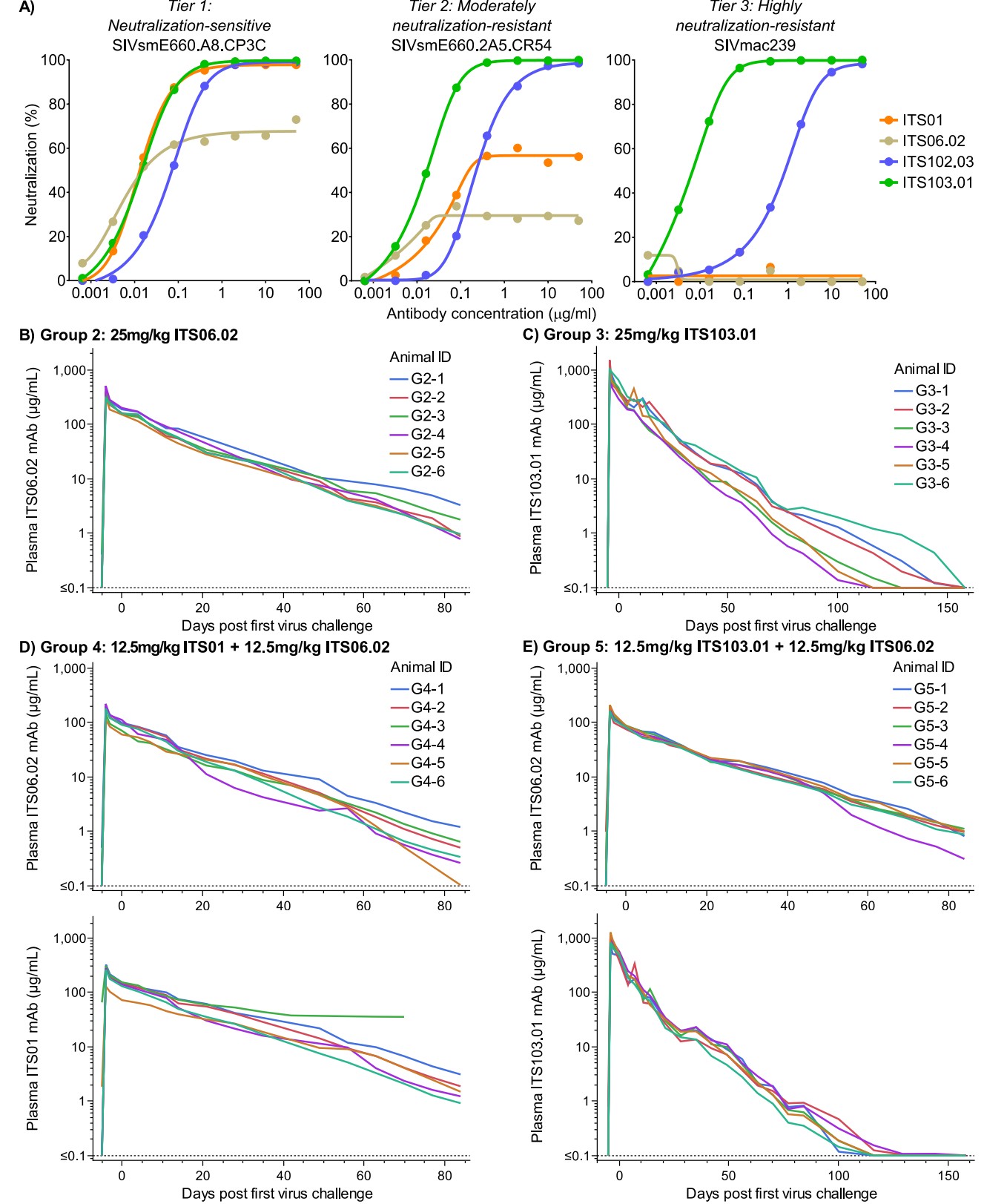

**Extended Data Fig. 1 | mAb neutralization against SIV isolates, and fully and partially neutralizing LS mAb pharmacokinetics in plasma.** (A) Neutralization activity of study mAbs against SIVsmE660.A8.CP3C (tier 1), SIVsmE660.2A5. CR54 (tier 2), and SIVmac239 (tier 3). Longitudinal plasma (B) ITS06.02, (C) ITS103.01, (D) ITS01 and ITS06.02, and (E) ITS103.01 and ITS06.02 mAb concentrations in animals from groups 2–5, respectively, from fully and partially neutralizing antibody infusion study relative to days following the first virus challenge. The dose of mAb used is indicated. (D–E) Separate graphs are shown for each mAb.

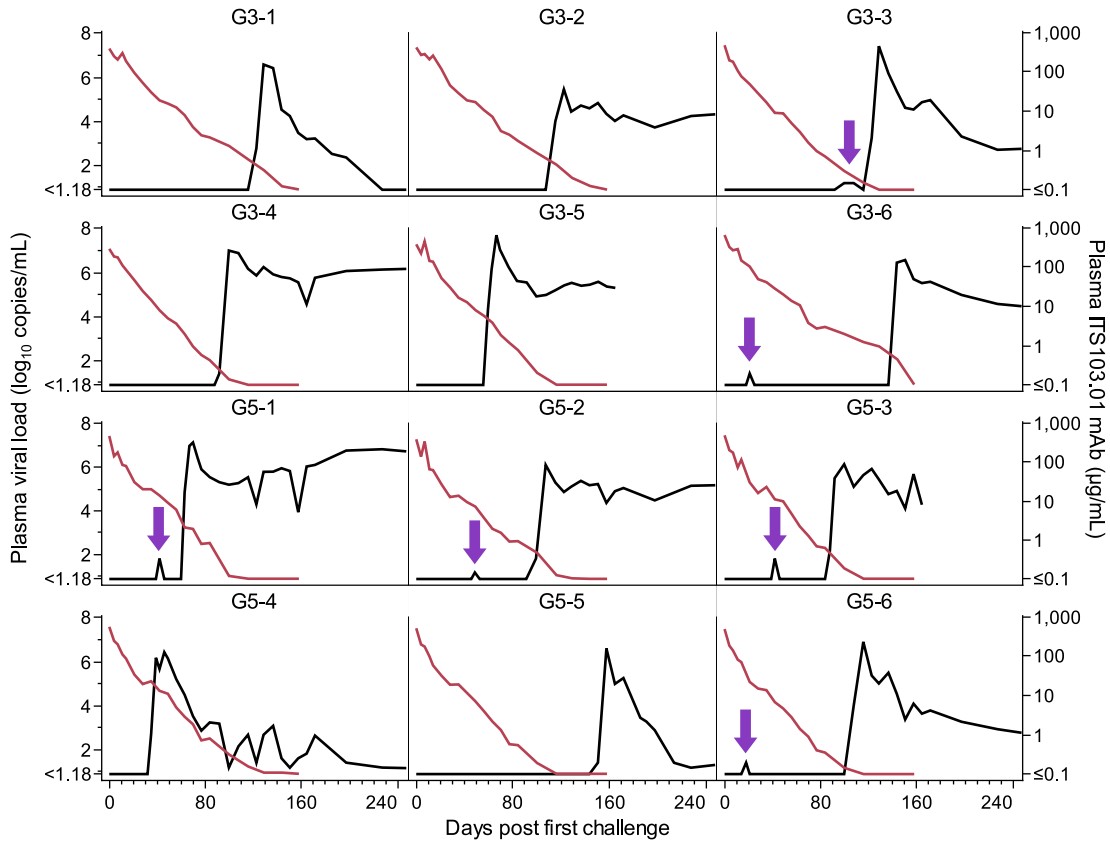

**Extended Data Fig. 2 | Viral blips in animals treated with ITS103.01 and challenged with SIVsmE660.FL14-IAKN.** Longitudinal plasma viral load (black line, left vertical axis) and plasma ITS103.01 mAb levels (red line, right vertical axis) up to 240 days post the first challenge are overlaid with each panel for an individual animal in groups 3 and 5 of the fully and partially neutralizing antibody infusion study. Viral blips (transient viremia of <1000 copies/mL plasma followed by ≥1 week of undetectable [<15 copies/mL] viral load) are indicated by the purple arrows.

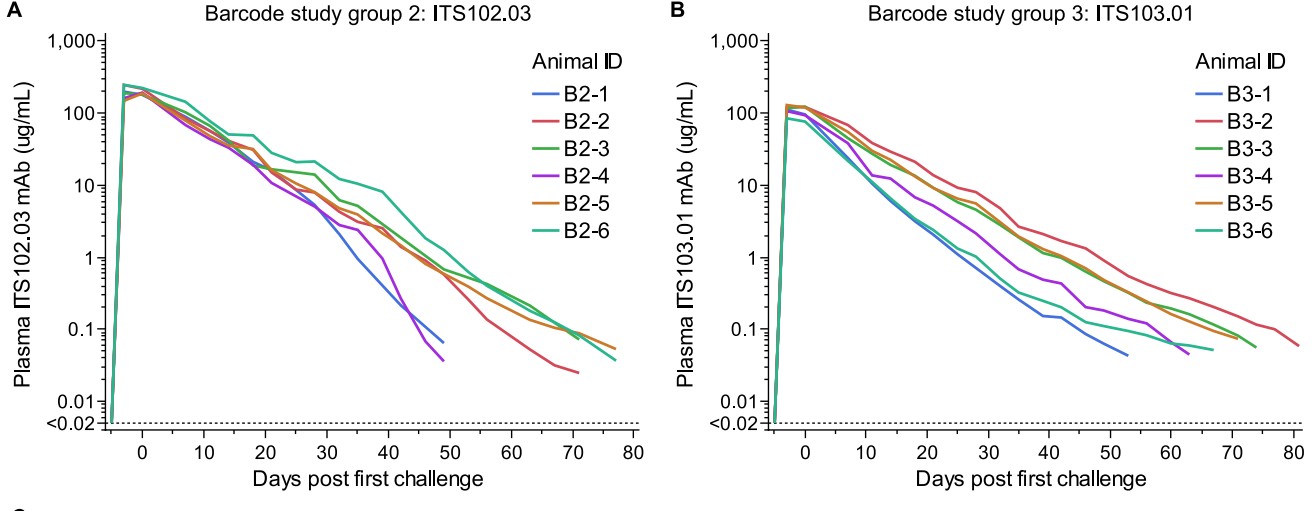

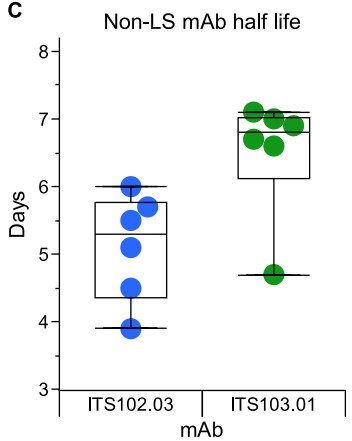

**Extended Data Fig. 3 | Non-LS ITS102.03 and ITS103.01 mAb pharmacokinetics in plasma.** Longitudinal plasma (A) ITS102.03 and (B) ITS103.01 mAb concentrations in animals from groups 2 and 3, respectively, of the barcoded SIV challenge study relative to days following the first virus challenge (mAbs infused at day -5). (C) Antibody half-life in days of infused non-LS mAbs used in barcoded SIV challenge study (individual animal values are plotted; n = 6 animals per mAb-treatment group). The overlaid boxes and middle line indicate the IQR and median, respectively, and the whiskers are the range.

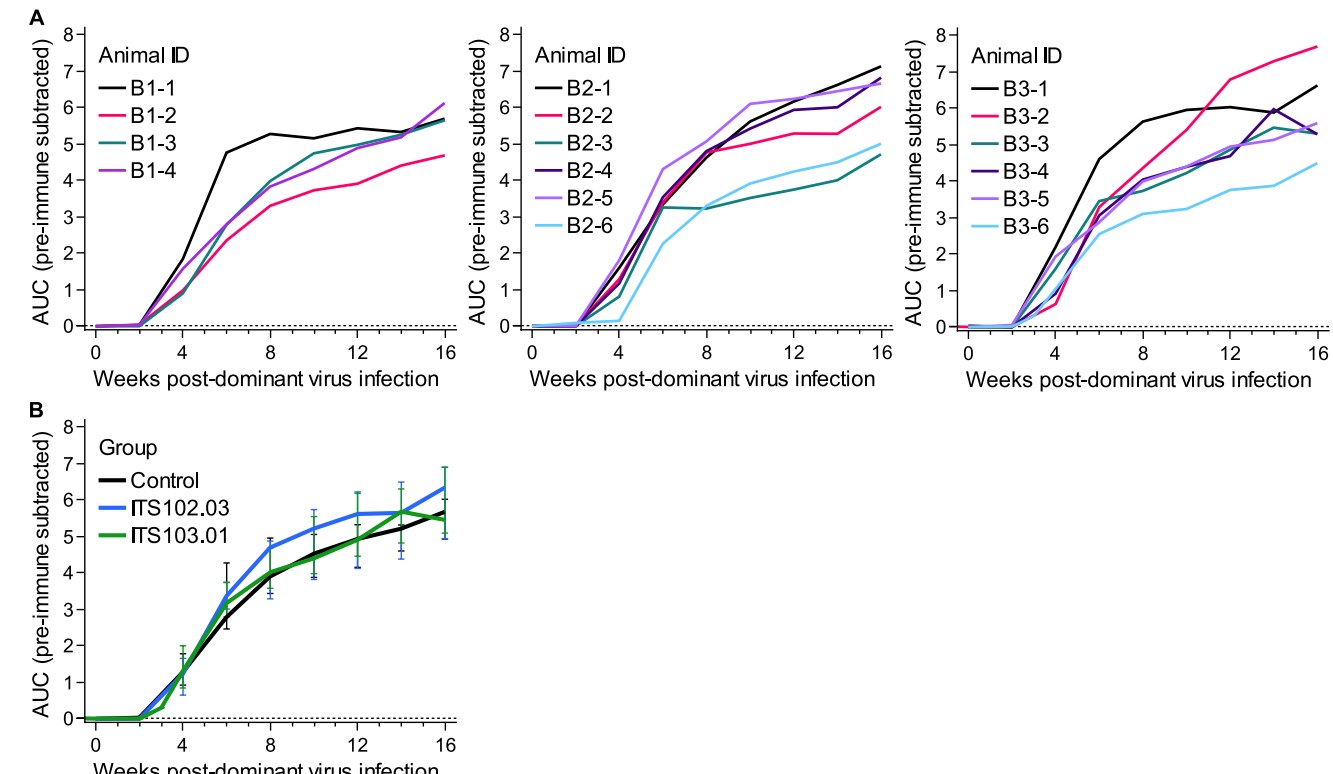

**Extended Data Fig. 4 | Endogenous SIV Env-specific humoral responses in barcoded SIV challenge study animals.** Longitudinal plasma antibody binding titers against SIVmac239 recombinant trimer measured as area under the curve (AUC) with pre-immune signal subtracted from 0 to 16 weeks post the dominant virus infection (as defined by the most abundant plasma virus barcode once sustained viremia occurred in each animal, see also supplementary Tables 1–3).

(A) Each graph shows individual animal responses from groups 1–3 (from left to right, respectively). (B) Each group's median antibody titer at each timepoint is shown with errors bars representing the IQR. Dotted lines equal 0. N = 4 (control group) or n = 6 (ITS102.03 and ITS103.1-treatment groups) animals.

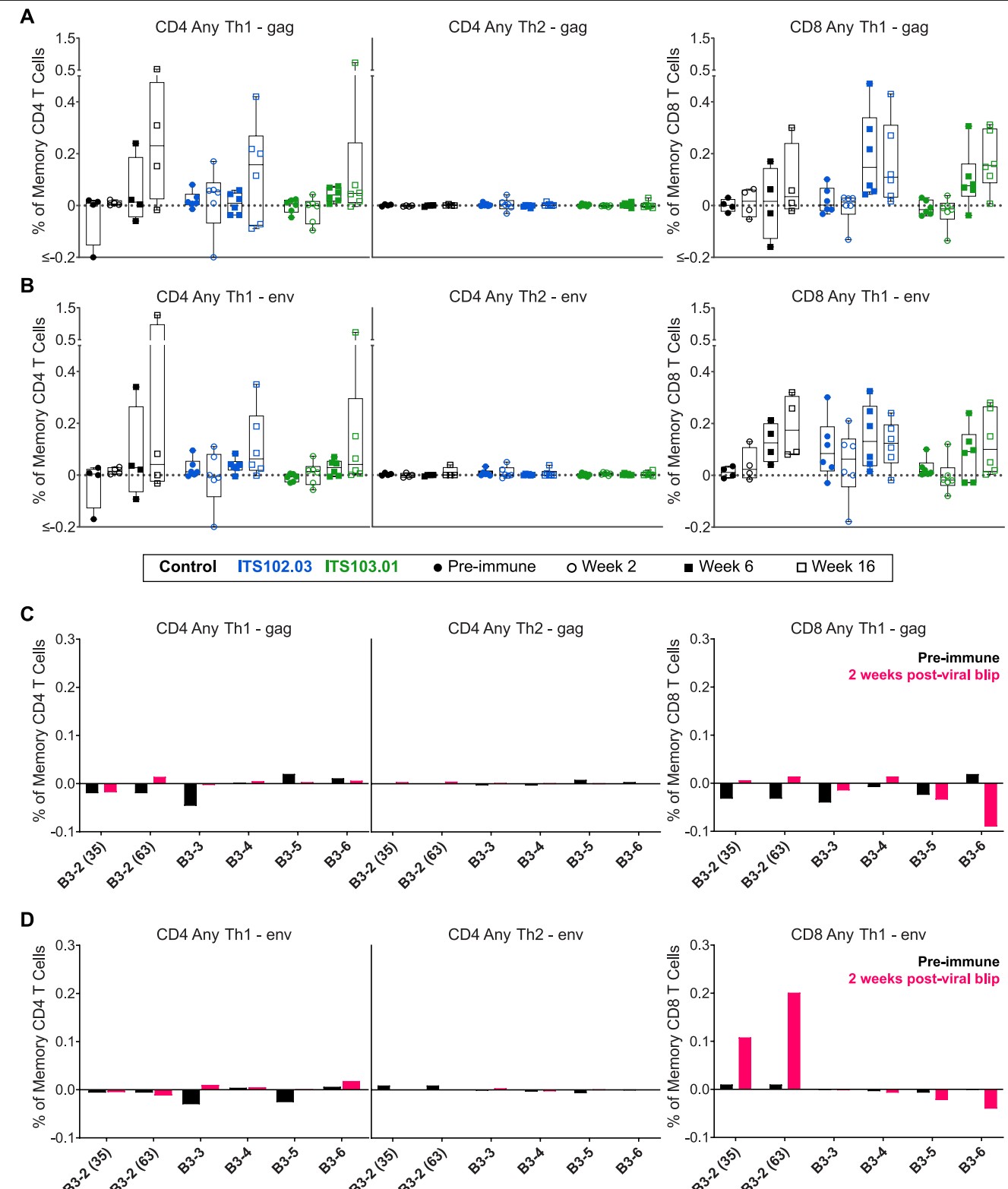

**Extended Data Fig. 5 | Endogenous SIV Gag- and Env-specific T cell responses in barcoded SIV challenge study animals.** Intracellular cytokine staining was performed on PBMCs to assess T cell responses to (A, C) SIV Gag and (B, D) Env peptide pools. (A, B) Samples assessed were collected before antibody infusion (pre-immune) and 2, 6 and 16 weeks post the dominant virus infection (see Tables S1–3). Circles and squares represent individual animals. Dotted lines are equal to 0%. The overlaid boxes and middle line indicate the IQR and median, respectively, and the whiskers are the range. N = 4 (control group) or n = 6 (ITS102.03 and ITS103.1-treatment groups) animals. (C, D) For animals where a viral blip was observed (animal IDs shown below horizontal axis and day post-first virus challenge of relevant blip in parentheses where appropriate), PBMCs collected 2 weeks after the blip were assayed and are shown relative to individual's pre-immune measurement. Solid, horizontal lines are equal to 0%. (A–D) CD4+ T cell Th1 responses (IFNγ, IL-2, or TNF), CD4+ T cell Th2 responses (IL-4 or IL-13), and CD8+ T cell responses (IFNγ, IL-2, or TNF) are shown from left to right, respectively. Responses are background-subtracted.

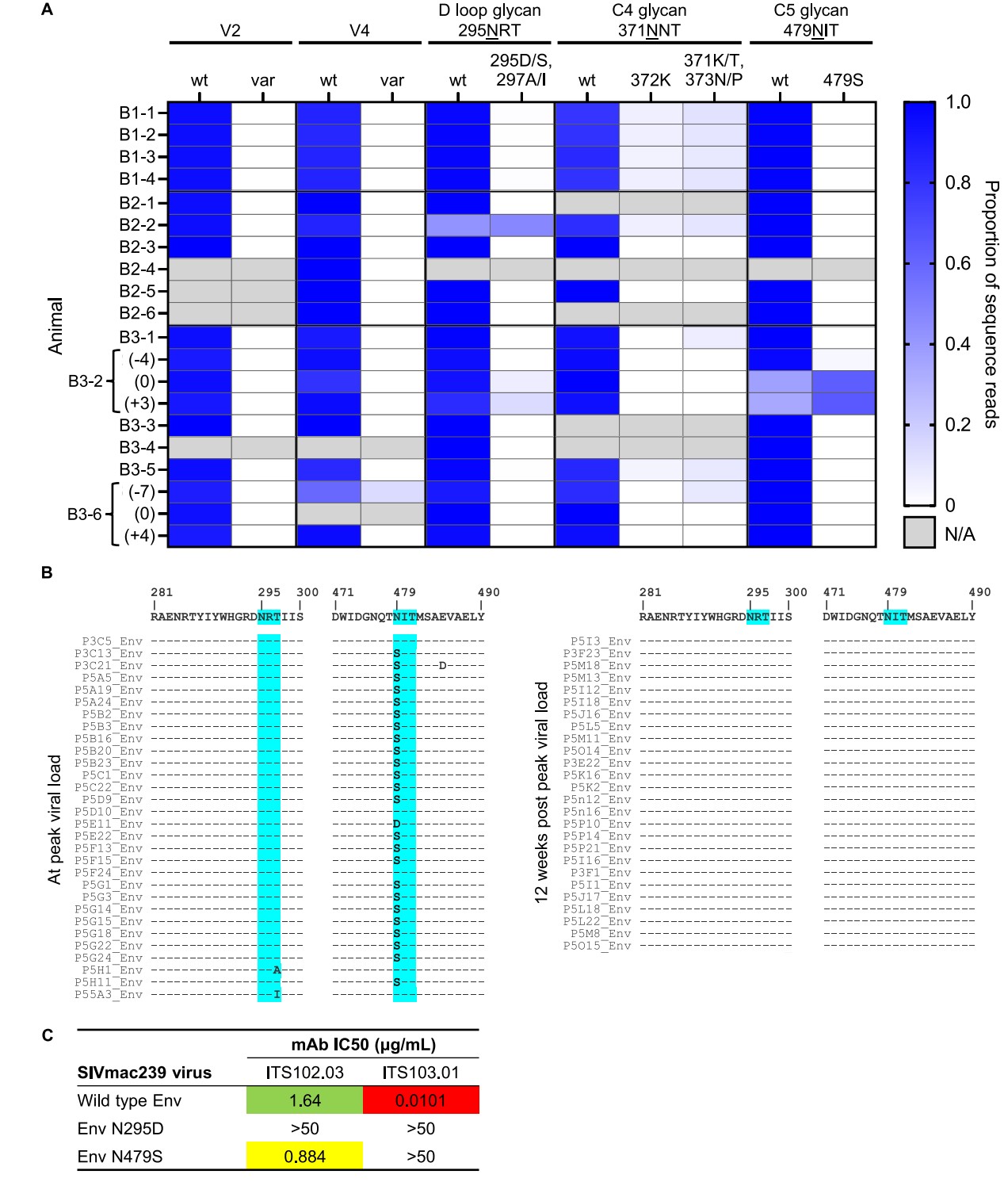

**Extended Data Fig. 6 | Env mutations detected in barcoded SIV challenge study animals.** (A) Sequencing at defined regions of Env as indicated along top of graph. Reads classified as not wild type (wt) are combined as general sequence variants (var) or as specific amino acid substitutions for PNGSs. Numbering according to HXB2 Env. All sequences are at peak viral load or additional timepoints as indicated (number of days +/− peak in parentheses). "N/A", not assessed. (B) Single genome reads for B3-2 at peak viral load and 12 weeks later. Each row is a unique clone. The sequence around C4 and C5 glycan sites is shown with reference sequence in bold above and the PNGS is highlighted. (C) Neutralization IC50s for ITS102.03 and ITS103.01 against wt and Env mutants.

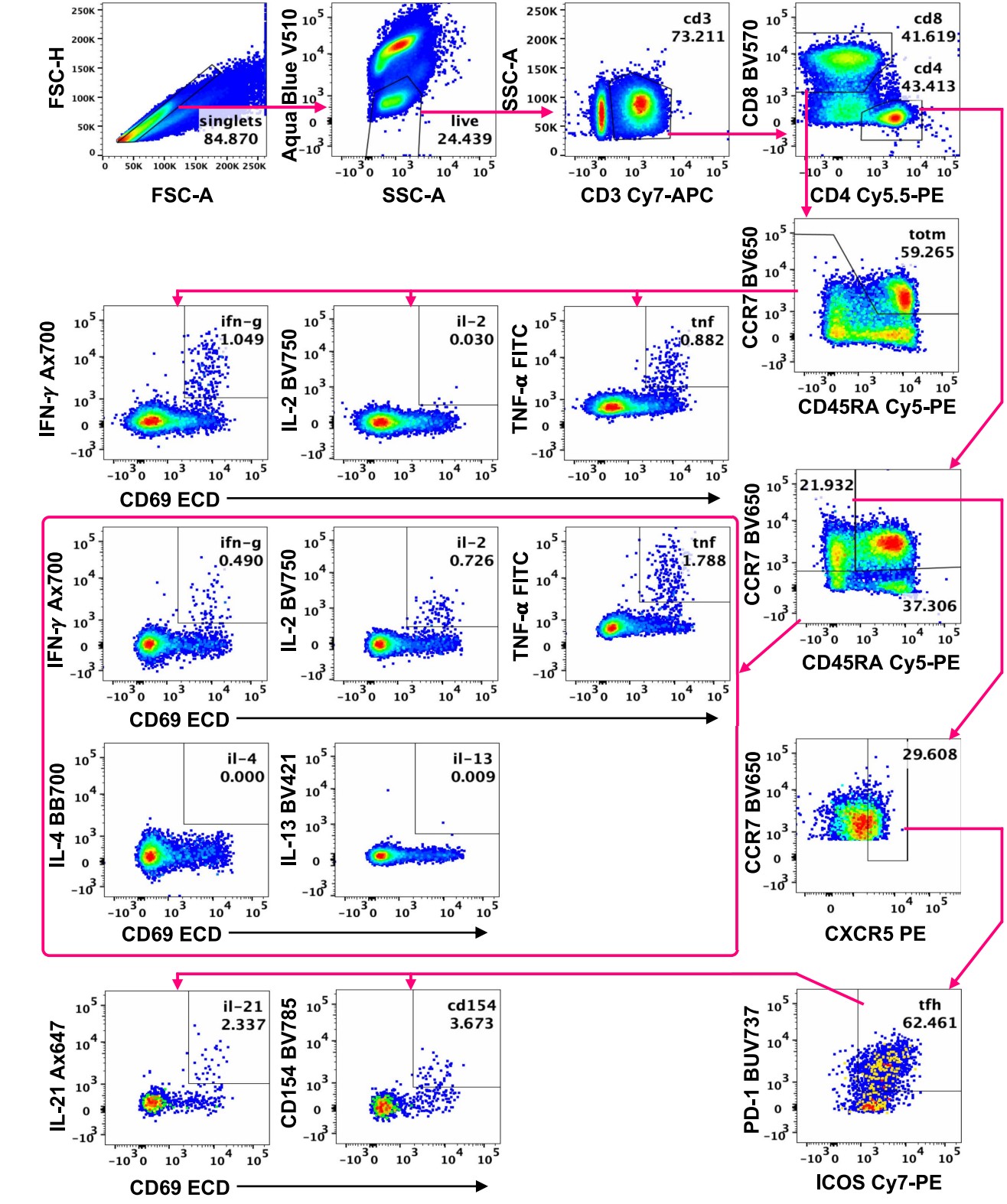

**Extended Data Fig. 7 | Gating strategy for intracellular cytokine staining.**
Flow cytometry gating strategy used to identify and quantify antigen-specific
T cells in PBMCs is shown from a representative sample. Sequential gating was
used to identify T cells, followed by memory CD4+ and CD8+ T cells. Cytokine-
expressing cells were then identified by co-expression of each cytokine with
upregulated CD69 expression.

# Reporting Summary

## Statistics

For all statistical analyses, confirm that the following items are present in the figure legend, table legend, main text, or Methods section.

| n/a | Confirmed | |
|---|---|---|
| ☐ | ☒ | The exact sample size (*n*) for each experimental group/condition, given as a discrete number and unit of measurement |
| ☐ | ☒ | A statement on whether measurements were taken from distinct samples or whether the same sample was measured repeatedly |
| ☐ | ☒ | The statistical test(s) used AND whether they are one- or two-sided<br>*Only common tests should be described solely by name; describe more complex techniques in the Methods section.* |
| ☒ | ☐ | A description of all covariates tested |
| ☐ | ☒ | A description of any assumptions or corrections, such as tests of normality and adjustment for multiple comparisons |
| ☐ | ☒ | A full description of the statistical parameters including central tendency (e.g. means) or other basic estimates (e.g. regression coefficient) AND variation (e.g. standard deviation) or associated estimates of uncertainty (e.g. confidence intervals) |
| ☐ | ☒ | For null hypothesis testing, the test statistic (e.g. *F*, *t*, *r*) with confidence intervals, effect sizes, degrees of freedom and *P* value noted<br>*Give P values as exact values whenever suitable.* |
| ☒ | ☐ | For Bayesian analysis, information on the choice of priors and Markov chain Monte Carlo settings |
| ☒ | ☐ | For hierarchical and complex designs, identification of the appropriate level for tests and full reporting of outcomes |
| ☒ | ☐ | Estimates of effect sizes (e.g. Cohen's *d*, Pearson's *r*), indicating how they were calculated |

*Our web collection on statistics for biologists contains articles on many of the points above.*

## Software and code

Policy information about availability of computer code

| | |
|---|---|
| Data collection | ELISA data was collected on a SpectraMax 384 Plus Absorbance Plate Reader (Molecular Devices)<br>Flow cytometry data was collected on a BD FACSymphony flow cytometer.<br>Neutralization data was collected on a SpectraMax L luminometer (Molecular Devices).<br>For single genome amplification, amplicons were Sanger sequenced on a 3730XL DNA Analyzer (Thermo Fischer Scientific).<br>Next-generation sequencing was performed using a MiSeq instrument (Illumina). |
| Data analysis | Data analysis was performed in GraphPad Prism v10.0.3 and JMP 15.<br>Flow cytometry data was analyzed using using FlowJo version 10.8.2 (Treestar, Inc., Ashland, OR).<br>Single genome amplification sequences were aligned using Geneious Prime software version 2023.0.<br>Sequence analysis was performed using the Frederick National Laboratory for Cancer Research Barcode Analysis Tool version 2022 (https://frederick.cancer.gov/research/aids-and-cancer-virus-program/sections/retroviral-evolution-section OR https://github.com/KeeleLab?tab=repositories), a custom algorithm written in R (version 4.3.1) to analyze barcoded viruses and MiSeq Env. |

For manuscripts utilizing custom algorithms or software that are central to the research but not yet described in published literature, software must be made available to editors and reviewers. We strongly encourage code deposition in a community repository (e.g. GitHub). See the Nature Portfolio guidelines for submitting code & software for further information.

# Data

Policy information about availability of data

All manuscripts must include a data availability statement. This statement should provide the following information, where applicable:
- Accession codes, unique identifiers, or web links for publicly available datasets
- A description of any restrictions on data availability
- For clinical datasets or third party data, please ensure that the statement adheres to our policy

The single genome sequence data have been deposited in GenBank with the accession codes PP458374 - PP458484 (http://www.ncbi.nlm.nih.gov/nuccore/). The MiSeq data have been deposited with links to BioProject accession number PRJNA1086015 in the NCBI BioProject database (https://www.ncbi.nlm.nih.gov/bioproject/). The other data that support the findings of this study are available in the source data document or from the corresponding author upon reasonable request.

# Research involving human participants, their data, or biological material

Policy information about studies with human participants or human data. See also policy information about sex, gender (identity/presentation), and sexual orientation and race, ethnicity and racism.

| | |
|---|---|
| Reporting on sex and gender | Not applicable |
| Reporting on race, ethnicity, or other socially relevant groupings | Not applicable |
| Population characteristics | Not applicable |
| Recruitment | Not applicable |
| Ethics oversight | Not applicable |

Note that full information on the approval of the study protocol must also be provided in the manuscript.

# Field-specific reporting

Please select the one below that is the best fit for your research. If you are not sure, read the appropriate sections before making your selection.

☒ Life sciences  ☐ Behavioural & social sciences  ☐ Ecological, evolutionary & environmental sciences

For a reference copy of the document with all sections, see nature.com/documents/nr-reporting-summary-flat.pdf

# Life sciences study design

All studies must disclose on these points even when the disclosure is negative.

| | |
|---|---|
| Sample size | Sample size was determined using power calculations for resolving differences between two means (independent samples t test) using a type I error rate of 0.05 and estimated average times to infection (in weeks) using historical data. For the first animal study, control animals were assumed to become infected on average at the first challenge (SD = 1) and mAb-treated animals would have a 2 week delay to infection (SD = 1). For a minimum of 85% power, 6 animals were required per group (power = 87.6%). Results from the first study were then used to inform estimates in power calculations for the second study. Using the relative difference in ITS103.01 and ITS102.03 in vitro IC80 values, 6 animals in each mAb infusion group was determined to provide >95% power to detect a difference in the time to infection between groups 2 and 3. For a minimum of 85% power, 4 animals in the control group were calculated to be sufficient (provides >95% or 91% power to detect a difference with ITS103.01 or ITS102.03 groups, respectively). |
| Data exclusions | Calculation of an ITS01 half-life for animal G4-3 is not shown due to high endogenous reactivity to the ITS01 anti-idiotypic antibody preventing calculation of mAb decay curve. |
| Replication | Animal studies were not replicated. Viral load measurements were generally not replicated given the limited amount of plasma available from the frequent blood sampling schedule utilized, however where practical samples were run in duplicate or triplicate. Plasma infused antibody measurement was repeated for a given sample at 2-3 different dilutions and an average value calculated. Neutralization and Env binding assays were performed in duplicate or triplicate. Intracellular cytokine staining was not replicated given the limited number of cells available at most time points. Sequencing was generally not replicated, except for repeated attempts at determining viral barcodes from viral blips. |
| Randomization | Animals were balanced into experimental groups on the basis of age, weight, sex, and (where required) TRIM5a genotype. The samples collected from these animals and used in the assays reported in the manuscript were allocated according to the animal/group from which they were derived. |
| Blinding | Investigators were not blinded to the treatments the animals received as they coordinated the study. Viral load testing and sequencing was performed with investigators blind to group allocation during data collection and initial analysis. Antibody measurements were not performed |

blind due to the substantial time and cost involved with assaying all animals for all potential infused mAbs at all timepoints. ICS was performed blind to the animal group allocation.

# Reporting for specific materials, systems and methods

We require information from authors about some types of materials, experimental systems and methods used in many studies. Here, indicate whether each material, system or method listed is relevant to your study. If you are not sure if a list item applies to your research, read the appropriate section before selecting a response.

## Materials & experimental systems

| n/a | Involved in the study |
|---|---|
| ☐ | ☒ Antibodies |
| ☐ | ☒ Eukaryotic cell lines |
| ☒ | ☐ Palaeontology and archaeology |
| ☐ | ☒ Animals and other organisms |
| ☒ | ☐ Clinical data |
| ☒ | ☐ Dual use research of concern |
| ☒ | ☐ Plants |

## Methods

| n/a | Involved in the study |
|---|---|
| ☒ | ☐ ChIP-seq |
| ☐ | ☒ Flow cytometry |
| ☒ | ☐ MRI-based neuroimaging |

## Antibodies

| Antibodies used | For intracellular cytokine staining (see OMIP referenced in methods section for concentrations used):<br>Antibody, Supplier, Clone number, Lot number<br>CD3 APC-Cy7, BD Biosciences, SP34.2, 1152687<br>CD4 PE-Cy5.5, Thermo Fisher, SK3, 2516573<br>CD8 BV570, BioLegend, RPA-T8, B346256<br>CD45RA PE-Cy5, BD Biosciences, 5H9, 8200578<br>CCR7 BV650, BioLegend, G043H7, B340645<br>CXCR5 PE, Thermo Fisher, MU5UBEE, 2404260<br>CXCR3 BV711, BD Biosciences, 1C6/CXCR3, 0309602<br>PD-1 BUV737, BD Biosciences, EH12.1, 0303349<br>ICOS Pe-Cy7, BioLegend, C398.4A, B293719<br>CD69 ECD, Beckman Coulter, TP1.55.3, 7620097<br>IFNg Ax700, BioLegend, B27, B320892<br>IL-2 BV750, BD Biosciences, MQ1-17H12, 2033660<br>IL-4 BB700, BD Biosciences, MP4-25D2, 1042139<br>TNF-FITC, BD Biosciences, Mab11, 1145433<br>IL-13 BV421, BD Biosciences, JES10-5A2, 1200672<br>IL-17 BV605, BioLegend, BL168, B338018<br>IL-21 Ax647, BD Biosciences, 3A3-N2.1, 1179052<br>CD154 BV785, BioLegend, 24-31, B329207<br><br>Anti-SIV mAbs:<br>Antibody, Supplier, Lot number<br>ITS01-LS, expressed in-house<br>ITS06.02-LS, expressed in-house<br>ITS103.01-LS, VPP, NTT627-160-04<br>ITS103.01, expressed in-house<br>ITS102.03, expressed in-house<br><br>Anti-idiotype mAbs:<br>Antibody, Supplier<br>17B4-IgG1 (anti-ITS01), expressed in-house<br>8A4-IgG1 (anti-ITS06.02), expressed in-house<br>anti-ITS103-Id1, expressed in-house<br>anti-ITS102-Id1, expressed in-house<br><br>Other antibodies:<br>Antibody, Supplier, Catalog No. (reactivity information)<br>Mouse Anti-Monkey IgG-HRP, SouthernBiotech, SB108a (Rhesus and cynomolgus IgG. Minimal reactivity to human and rabbit immunoglobulins, goat IgG, and mouse, rat, hamster, guinea pig, sheep, donkey, bovine, horse, porcine, feline, and chicken serum)<br>Goat Anti-Human Ig κ chain-HRP, Millipore Sigma, AP502P (Human. Reacts with human kappa light chains. Absorbed for human myeloma proteins with λ light chains and mouse immunoglobulins) |
|---|---|
| Validation | The ITS01, ITS06.02, ITS103.01, ITS102.03 mAbs were described in the manuscripts Mason et al, PLoS Pathogens 2016, PMID 27064278 and Welles et al, PLoS Pathogens 2022, PMID 35709309.<br>The ITS103.01-LS mAb was biochemically validated by the NIH VRC Vaccine Production Program. All ITS mAbs used for administration (ITS01-LS, ITS06.02-LS, ITS103.01-LS, ITS103.01, ITS102.03) were validated for purity by SDS-PAGE, antigen binding by ELISA assay, and activity by neutralization assay. |

The anti-idiotype antibodies were described in the manuscripts Welles et al, PLoS Pathogens 2018, PMID 30517201 and Welles et al, PLoS Pathogens 2022, PMID 35709309 and validated by binding to their cognate antibody.
All other antibodies were commercially available products and subjected to routine testing by the supplying vendor.

# Eukaryotic cell lines

Policy information about cell lines and Sex and Gender in Research

| | |
|---|---|
| Cell line source(s) | 293T (human cell line), female, ATCC<br>Expi293F (human cell line), female, ThermoFisher Scientific<br>TZM-bl (human cell line), female, NIH HIV Reagent Program |
| Authentication | Cell lines were all obtained from commercial sources/repositiories and were authenticated by the organizations from which they were obtained, specifically:<br>293T (ATCC) - STR profiling<br>Expi293F (ThermoFisher Scientific) - STR profiling and assessment of cell morphology/growth kinetics<br>TZM-bl (NIH HIV Reagent Program) - assessment of cell morphology and functional testing in neutralization assays |
| Mycoplasma contamination | Cell lines were not tested for mycoplasma contamination. |
| Commonly misidentified lines<br>(See ICLAC register) | No commonly misidentified cell lines were used in this study. |

# Animals and other research organisms

Policy information about studies involving animals; ARRIVE guidelines recommended for reporting animal research, and Sex and Gender in Research

| | |
|---|---|
| Laboratory animals | Forty-six research naïve, Indian origin rhesus macaques were used, aged between 2-5 years and weighing between 3-10kg. |
| Wild animals | No wild animals were used in this study. |
| Reporting on sex | In the first animal study, there were 7 females and 23 males. The females were distributed across the study groups as part of the randomization process. The second study used all male animals. Sex-based analysis was not performed due to the small study sample size. |
| Field-collected samples | No field-collected samples were used in this study. |
| Ethics oversight | All experiments were carried out in compliance with National Institutes of Health regulations and approval from the Animal Care and Use Committee of the Vaccine Research Center and Bioqual, Inc (Rockville, MD, USA), where NHPs were housed for the duration of the studies. Animals were housed and cared for in accordance with local, state, federal and institutional policies in facilities accredited by AAALAC International under standards established in the Animal Welfare Act and the Guide for the Care and Use of Laboratory Animals. In accordance with the institutional policies of both institutions, all compatible non-human primates are always pair-housed, and single housing is only permissible when scientifically justified or for veterinary medical reasons, and for the shortest duration possible. Non-human primates were housed in appropriately sized caging according to the Guide for the Care and Use of Laboratory Animals, eighth ed.40, and supplemented with a variety of enrichment toys, treats, fresh produce, and foraging devices. Water was offered ad libitum and animals were fed primate biscuits (Monkey Diet, 5038, Lab diet) twice daily. As standard practice, animal holding rooms were maintained on a 12-hour light/dark cycle, room temperature of 16–21°C, and relative humidity 30–70%. |

Note that full information on the approval of the study protocol must also be provided in the manuscript.

# Plants

| | |
|---|---|
| Seed stocks | *Report on the source of all seed stocks or other plant material used. If applicable, state the seed stock centre and catalogue number. If plant specimens were collected from the field, describe the collection location, date and sampling procedures.* |
| Novel plant genotypes | *Describe the methods by which all novel plant genotypes were produced. This includes those generated by transgenic approaches, gene editing, chemical/radiation-based mutagenesis and hybridization. For transgenic lines, describe the transformation method, the number of independent lines analyzed and the generation upon which experiments were performed. For gene-edited lines, describe the editor used, the endogenous sequence targeted for editing, the targeting guide RNA sequence (if applicable) and how the editor was applied.* |
| Authentication | *Describe any authentication procedures for each seed stock used or novel genotype generated. Describe any experiments used to assess the effect of a mutation and, where applicable, how potential secondary effects (e.g. second site T-DNA insertions, mosiacism, off-target gene editing) were examined.* |

# Flow Cytometry

## Plots

Confirm that:

☒ The axis labels state the marker and fluorochrome used (e.g. CD4-FITC).

☒ The axis scales are clearly visible. Include numbers along axes only for bottom left plot of group (a 'group' is an analysis of identical markers).

☒ All plots are contour plots with outliers or pseudocolor plots.

☒ A numerical value for number of cells or percentage (with statistics) is provided.

## Methodology

| | |
|---|---|
| Sample preparation | Cryopreserved PBMC from rhesus macaques were used for flow cytometry |
| Instrument | BD FACSymphony flow cytometer |
| Software | FlowJo version 10.8.2 (Treestar, Inc., Ashland, OR) |
| Cell population abundance | No cell sorting was performed |
| Gating strategy | For intracellular cytokine staining, single cells were identified based on FSC-A vs FSC-H, then live lymphocytes were identified using SSC-A vs Aqua blue viability stain. Cell were then gated as being CD3+, CD4 or CD8+, then as memory cells based on CD45RA- CCR7-, or CD45RA-CCR7+ or CD45RA+CCR7- prior to identifying cytokine positive cells based upon co-expression of the cytokine and CD69. Tfh-like cells were identified as CCR7+ CXCR5+ and PD-1+ ICOS+. |

☒ Tick this box to confirm that a figure exemplifying the gating strategy is provided in the Supplementary Information.

