## [Peer Review File · Nature]

Antibody prophylaxis may mask subclinical SIV infections in macaques

Corresponding Author: Dr Mario Roederer

Version 0:

Reviewer comments:

Referee #1

(Remarks to the Author)

The manuscript by Gonelli et al. describes studies conducted in rhesus macaques to evaluate and define levels of passively infused rhesus-derived bNAbs that provide protection against breakthrough infection following weekly intra-rectal challenges with Tier 2 or Tier 3 SIV challenge viruses. Using single or combinations of rhesus bNAbs that are defined as “partially” or “fully” neutralizing, the authors demonstrate that only monkeys receiving the most potent fully neutralizing mAb ITS103 (anti-CD4bs) demonstrate a significant delay in time to infection, whereas partially neutralizing mAbs had minimal if any impact. Once breakthrough infection occurred, the kinetics of viral replication and peak/setpoint viral loads were similar in mAb treated and control animals. To investigate viral blips that were observed in ITS103 treated animals prior to breakthrough infection, an elegant strategy of employing 8 different genetically barcoded SIVmac239 challenge stocks was used to define which weekly challenge resulted in subclinical infections and/or onset of acute viremia. In 5/6 animals there was evidence that viral challenges prior to the dominant viral infection (ranging from 1 to 8 weeks) had caused infection but did not result in viremia and only became detectable when bnAb levels fell below protective levels. One limitation of the study was the inability to confirm the identity of viruses causing the viral blips in 4 of 5 monkeys, although barcoding did confirm and identify the presence of subclinical infections prior to breakthrough infection. Subclinical infections occurred when bnAb levels were ~98-239-fold above IC80 levels, and breakthrough infection occurred when bnAb levels fell to ~12-14-fold above IC80. Analysis of autologous antibody and T cell immunity post-infection did not demonstrate differences between mAb treated and control animals, and only a few monkeys had defined viral bnAb antibody escape mutations that were transient in nature. The authors conclude that the well-controlled SIV model in rhesus macaques provides evidence of subclinical infections occurring in the presence of high serum levels of bnAb and allows for better definition of bnAb levels required for protection. Another key observation was the variability in prevention levels observed between individual animals which the authors argue may impact interpretation of human bnAb clinical trials.

Overall this was a well conducted study that provides important insight into bnAb levels required for protection against subclinical and breakthrough infections. While there may exist differences in the NHP/SIV model and HIV-1 infection in humans, this study remains informative for the field in regards to protective levels of bnAb required for prevention of acute viremia and outlines critical parameters that are more difficult to assess in large human trials. The manuscript is well written, the data are presented in clear figures and tables, and appropriate statistics were used for analysis. Below are additional comments for consideration.

- 1) The authors consistently use the terms “fully” or “partially” neutralizing when describing the mAbs used in this study, but there is no clear description of what those terms define. Are partially neutralizing antibodies less potent, exhibit incomplete neutralization curves, or other? What are the IC80 titers of ITS01 and ITS06 against SIVsmE660? This needs to be clarified, and it may be useful to include raw neutralization curves in the extended data for each mAb against the SIVmacE660 and SIVmac239 challenge viruses.
- 2) Did the authors assess whether partially neutralizing antibodies selected for higher levels of escape mutations in those targeted epitopes?
- 3) The study relied on measurement of mAb levels in plasma to correlate with protection, which is consistent with human bnAb trials. Do the authors have data demonstrating that all bnAbs used in this study exhibit similar levels of diffusion in rectal mucosal tissue following passive administration?
- 4) Line 139: reference to Figure 3D is missing.
- 5) Line 427: the cross-reference to Extended Fig. 1 A and C here doesn't make sense.

Referee #2

(Remarks to the Author)

Gonelli and colleagues present a well-designed and well-written report testing antibody immunoprophylaxis in the SIV/NHP model. Using two distinct barcoded challenge virus systems and well-characterized anti-SIV mAbs, they demonstrate protection with only fully neutralizing mAbs, model the impact of the various mAbs on virus kinetics of acquisition, and raise provocative questions about detection of sub-clinical low-level viremia.

The novelty and robustness of the model systems (barcoded virus stocks, well-characterized SIV mAbs) are high, enabling detection of these low-level viremic episodes that have not been identified either in prior NHP systems or in human trials or cohorts.

My main concern with the authors' conclusions is in the framing of the finding of low-level viremia detected as sub-clinical infection that is masked by mAbs. Given the fact that the detected replicating barcoded-viruses were generally not detected subsequently within the populations of sustained replicating viruses (with one notable exception), nor were these sub-clinical viruses shown to have lasting impact in a context relevant to PWH (eg, detected within the reservoir on ART (provirus found during ART suppression) or within tissues at necropsy, or in a model where challenge leads to sub-clinical viremia, but not productive infection), it is hard to know if these represent productive infection, or a snuffed-out infection process that likely happens, but is unmeasured/undetected in various other NHP models or human exposures. Thus, it is appropriate for the authors to be cautious in their interpretation of this finding. They are cautious for the most part (eg, discussion lines 272-275), with the exception of the provocative title and the use of the word 'masks.'

Suggested improvements:

1. The authors appropriately suggest future study of low-level viremia in HIV-1 in humans. Additionally, further study within the NHP model could be performed. Within this experimental system, sequencing of provirus from PBMC longitudinally in RM with early low-level viremia of a distinct barcode from sustained viremia may provide an opportunity to determine the extent of lasting infection within these RM. Future experiments where RM undergo mucosal challenge only in the presence of high-level bNAbs (as opposed to continuing challenges as levels wane to eventually sub-suppressive levels) to see if the detected "blips" translate to productive infection would be highly informative, but would require additional experiments of substantial time and cost.
2. While these data are provided later in the results, it would be helpful context to provide very early in the main text the IC80 of the challenge viruses to the various mAbs. This supports the otherwise qualitative (eg, weak) descriptions of the various mAbs. In addition, maximal percent inhibition of the mAbs per virus would be helpful.
3. Similarly, the expected half-lives of the administered mAbs should be stated initially, and the PK curves of the mAbs—either of each RM or a mean/median value should be a component of the main figures, as it helps provide context to the timing virus acquisition.
4. The use of person-first and de-stigmatizing language is generally excellent throughout the manuscript. The authors should consider the term "at-risk of HIV acquisition" as DAIDs and community groups have identified this as less ideal.
5. The viral and mAb features in Figure 2 may be better characterized with an annotated figure of the median +/- range of VL and mAb level, and smaller call out of the comparison between groups of the features in Figs 2 A-D.
6. The "no apparent vaccinal effect from infusion of mAb" section seems to discuss a lack of immunologic response to viral blips, more than to the mAb. In addition, the change in autologous neutralizing antibody responses may be virus-specific, and far more measurable against the relatively sensitive SIVsmE660 vs. SIVmac239. Autologous nAbs against the latter virus are often delayed or do not occur during the timeframe in which these responses almost universally arise in PWH. Similarly, the impact of the viruses on vaccinal effect should be mentioned in discussion lines 309-310.
7. The sentence at line 264-266 is ambiguous and should be clarified.

Referee #3

(Remarks to the Author)

Gonelli et al studied breakthrough SIV during the waning phase after administration of neutralising antibodies in macaques. They nicely show with twice weekly monitoring that viral blips are commonly seen prior to full infection. In the instance where the blip could be genotyped it corresponded to a barcoded SIV strain administered 1-2 weeks prior to the blip. The work is well done, has implications to Nab prevention of HIV in humans and should be of interest to Nature readers.

Comments

The viral blips are key to the interpretation of the work. In humans, sensitive viral load assays have been reported to be occasionally be weakly positive in the absence of HIV infection although the veracity of such claims is admittedly unclear. It is also very curious and somewhat worrying that in the second study 4 of the 5 blips detected were at the same time point (day 35) – this seems beyond the boundaries of a statistical fluke? The authors may wish to consider rigorously validating the blips by (a) doing VL tests on a large number of monkeys not exposed to SIV to show that the ~10 blips of ~200 negative VL samples are not spurious, (b) repeating the assays of the blips to show they are reproducible (c) trying harder to sequence the small amount of RNA in the blips - it is worrying that only one sample could be sequenced – in HIV it is common to get a positive sequencing result when the VL approaches 10e3, (d) looking for virus in CD4 T cell DNA (and sequencing if found).

Was the “full” vs “partial” neutralization comparison in Fig 1 and 2 a pre-specified endpoint with an analysis plan or a somewhat ad hoc analysis based on the data – if the latter it would be better to analyse each of the 5 groups separately. Fig 1E is not needed.

In Fig 1B, there seem to be more treated animals that ended up with low set point VLs (one or 2 per group). Was this driven by MHC genetics?

Although this work is highly novel, a similar anecdotal phenomenon was described in pigtailed macaques exposed to cell-associated SHIV a few years back with speculation on its implications in a follow up perspective doi: 10.1016/j.it.2017.12.006 doi: 10.1126/scitranslmed.aaf1483. That work speculated that suppressed infections occurring during Bnab treatment could recrudescence later. In this work, the continual SIV challenges made that hard to assess. An interesting follow up study would of course be to stop challenging once a blip is observed.

Truvada is widely used as pre-exposure prophylaxis and there have been reports of breakthrough HIV infections with some similarities to the blips reported here. Longer acting injectible antiretrovirals are now also being used for Prep. I think this work has potential implications for that field.

Version 1:

Reviewer comments:

Referee #1

(Remarks to the Author)

The authors have adequately responded to reviewer comments and made appropriate edits.

Referee #2

(Remarks to the Author)

The authors appropriately responded to my review, including a modification of the manuscript title (not reflected on this page).

Referee #3

(Remarks to the Author)

I am satisfied with the response.

Open Access This Peer Review File is licensed under a Creative Commons Attribution 4.0 International License, which permits use, sharing, adaptation, distribution and reproduction in any medium or format, as long as you give appropriate credit to the original author(s) and the source, provide a link to the Creative Commons license, and indicate if changes were

made.

The reviewers' comments/questions are presented in normal, black text while our responses are in bold, blue text. Line numbers in our responses are with respect to the revised manuscript without all markup shown (i.e. "simple markup").

Referee #1 (Remarks to the Author):

The manuscript by Gonelli et al. describes studies conducted in rhesus macaques to evaluate and define levels of passively infused rhesus-derived bNAbs that provide protection against breakthrough infection following weekly intra-rectal challenges with Tier 2 or Tier 3 SIV challenge viruses. Using single or combinations of rhesus bNAbs that are defined as "partially" or "fully" neutralizing, the authors demonstrate that only monkeys receiving the most potent fully neutralizing mAb ITS103 (anti-CD4bs) demonstrate a significant delay in time to infection, whereas partially neutralizing mAbs had minimal if any impact. Once breakthrough infection occurred, the kinetics of viral replication and peak/setpoint viral loads were similar in mAb treated and control animals. To investigate viral blips that were observed in ITS103 treated animals prior to breakthrough infection, an elegant strategy of employing 8 different genetically barcoded SIVmac239 challenge stocks was used to define which weekly challenge resulted in subclinical infections and/or onset of acute viremia. In 5/6 animals there was evidence that viral challenges prior to the dominant viral infection (ranging from 1 to 8 weeks) had caused infection but did not result in viremia and only became detectable when bnAb levels fell below protective levels. One limitation of the study was the inability to confirm the identity of viruses causing the viral blips in 4 of 5 monkeys, although barcoding did confirm and identify the presence of subclinical infections prior to breakthrough infection.

Subclinical infections occurred when bnAb levels were ~98-239-fold above IC80 levels, and breakthrough infection occurred when bnAb levels fell to ~12-14-fold above IC80. Analysis of autologous antibody and T cell immunity post-infection did not demonstrate differences between mAb treated and control animals, and only a few monkeys had defined viral bnAb antibody escape mutations that were transient in nature. The authors conclude that the well-controlled SIV model in rhesus macaques provides evidence of subclinical infections occurring in the presence of high serum levels of bnAb and allows for better definition of bnAb levels required for protection. Another key observation was the variability in prevention levels observed between individual animals which the authors argue may impact interpretation of human bnAb clinical trials.

Overall this was a well conducted study that provides important insight into bnAb levels required for protection against subclinical and breakthrough infections. While there may exist differences in the NHP/SIV model and HIV-1 infection in humans, this study remains informative for the field in regards to protective levels of bnAb required for prevention of acute viremia and outlines critical parameters that are more difficult to assess in large human trials. The manuscript is well written, the data are presented in clear figures and tables, and appropriate statistics were used for analysis. Below are additional comments for consideration.

We thank the reviewer for their comments and suggestions. In relation to their point about the failure to determine the viral barcode of 4/5 viral blips, we added text pointing this out in the discussion. However, the low copy barcode reads observed after sustained viremia do indicate that infections occurred while mAb levels were high and points to the blips being real.

1) The authors consistently use the terms "fully" or "partially" neutralizing when describing the mAbs used in this study, but there is no clear description of what those terms define. Are partially neutralizing antibodies less potent, exhibit incomplete neutralization curves, or other? What are the IC80 titers of ITS01 and ITS06 against SIVsmE660? This needs to be clarified, and it may be useful to include raw neutralization curves in the extended data for each mAb against the SIVmacE660 and SIVmac239 challenge viruses.

We thank the reviewer for highlighting the poor definition of mAb neutralization behavior. The neutralization curves for the mAbs were added to extended figure 1 as well as a table detailing the inferred IC50 and IC80 titers (new supplementary table 1). The results text was also amended to better detail these characteristics (line 67-74).

2) Did the authors assess whether partially neutralizing antibodies selected for higher levels of escape mutations in those targeted epitopes?

We have not conducted specific studies to assess if the partially/incompletely neutralizing antibodies select for escape mutations. However, escape mutations are not expected given the virus can replicate in the presence of these antibodies (i.e. no selective pressure). Even when there was pressure to escape antibody binding—as was the case for the completely neutralizing antibodies ITS103.01 and ITS102.03—significant mutations were only observed in 1 out of 6 treated animals and were transient as the infused mAb levels waned.

3) The study relied on measurement of mAb levels in plasma to correlate with protection, which is consistent with human bnAb trials. Do the authors have data demonstrating that all bnAbs used in this study exhibit similar levels of diffusion in rectal mucosal tissue following passive administration?

We do not have experimental data for the rectal mucosa translocation of mAbs used here. However, within each study, all the mAbs shared identical constant region sequences (rhesus macaque IgG1 with the LS mutation in the first study and wild-type IgG1 sequence in the barcoded virus study), so we expect the same level of tissue translocation. Additionally, previous analysis of antibody distribution in tissues has found that multiple mAbs translocate to tissues with respect to their plasma concentration within a 2-fold margin (Shah, Dhaval K, and Alison M Betts. *mAbs* vol. 5,2 (2013): 297-305).

4) Line 139: reference to Figure 3D is missing.

The figure reference has been added (line 145).

5) Line 427: the cross-reference to Extended Fig. 1 A and C here doesn't make sense.

The cross-reference was placed in error and has been removed.

Referee #2 (Remarks to the Author):

Gonelli and colleagues present a well-designed and well-written report testing antibody immunoprophylaxis in the SIV/NHP model. Using two distinct barcoded challenge virus systems and well-characterized anti-SIV mAbs, they demonstrate protection with only fully neutralizing mAbs, model the impact of the various mAbs on virus kinetics of acquisition, and raise provocative questions about detection of sub-clinical low-level viremia.

The novelty and robustness of the model systems (barcoded virus stocks, well-characterized SIV mAbs) are high, enabling detection of these low-level viremic episodes that have not been identified either in prior NHP systems or in human trials or cohorts.

My main concern with the authors' conclusions is in the framing of the finding of low-level viremia detected as sub-clinical infection that is masked by mAbs. Given the fact that the detected replicating barcoded-viruses were generally not detected subsequently within the populations of sustained replicating viruses (with one notable exception), nor were these sub-clinical viruses shown to have lasting impact in a context relevant to PWH (eg, detected within the reservoir on ART (provirus found during ART suppression) or within tissues at necropsy, or in a model where challenge leads to sub-clinical viremia, but not productive infection), it is hard to know if these represent productive infection, or a snuffed-out infection process that likely happens, but is unmeasured/undetected in various other NHP models or human exposures. Thus, it is appropriate for the authors to be cautious in their interpretation of this finding. They are cautious for the most part (eg, discussion lines 272-275), with the exception of the provocative title and the use of the word 'masks.'

We agree that the experimental approach employed in this paper does not determine if the subclinical infections (identified by barcode sequencing) can develop into a typical, productive infection. Further studies are underway to hopefully address this question and is beyond the scope of this paper. It should be noted that there were viruses with a barcode indicating they were the result of subclinical infections identified within the population of sustained replicating viruses, albeit at low frequency. Their low prevalence amongst the sustained replicating viruses is not unexpected since challenges administered after the infection that later became the dominant replicating virus often failed (in 11/12 mAb-treated animals) to appear as a significant proportion of the replicating virus (i.e. secondary infections did not readily take hold). Virus from a subclinical infection likely experiences that same limitations in its ability to expand; once the neutralizing mAb levels decrease sufficiently to not prevent virus release from (a likely small number) of infected cells and/or infection of new targets, an inoculum of free virus from an i.r. challenge will be able to infect many targets and expand exponentially faster. Nonetheless, we agree that the title is too provocative and qualified it (see line 1).

Suggested improvements:

1. The authors appropriately suggest future study of low-level viremia in HIV-1 in humans. Additionally, further study within the NHP model could be performed. Within this experimental system, sequencing of provirus from PBMC longitudinally in RM with early low-level viremia of a distinct barcode from sustained viremia may provide an opportunity to determine the extent of lasting infection within these RM. Future experiments where RM undergo mucosal challenge only in the presence of high-level bNAbs (as opposed to continuing challenges as levels wane to eventually sub-suppressive levels) to see if the detected "blips" translate to productive infection would be highly informative, but would require additional experiments of substantial time and cost. **We agree with the reviewer that additional study with the NHP/SIV model is warranted, and we have initiated studies along these lines. Those studies will require a large number of animals to complete and we are considering the appropriate design.** **Regarding the current experimental system and suggestion of sequencing provirus from PBMCs to look for viral barcodes associated with subclinical infections; based on past experience sequencing proviral DNA, we do not expect to reliably find evidence of proviral DNA in the samples collected longitudinally given the low viral loads observed at viral blips (if any) and the relatively small number of**

cells (~5 million PBMCs) collected at each timepoint (twice weekly sampling limited the blood volume we could safely collect).

2. While these data are provided later in the results, it would be helpful context to provide very early in the main text the IC80 of the challenge viruses to the various mAbs. This supports the otherwise qualitative (eg, weak) descriptions of the various mAbs. In addition, maximal percent inhibition of the mAbs per virus would be helpful.

We thank the reviewer for highlighting a lack of clear definition of the neutralizing activity of the mAbs used. The neutralization curves for the mAbs were added to extended figure 1 as well as a table detailing the inferred IC50 and IC80 titers (new supplementary table 1). At the start of the results, the text also details the inhibitory concentrations of the mAbs where applicable (lines 67-74).

3. Similarly, the expected half-lives of the administered mAbs should be stated initially, and the PK curves of the mAbs- either of each RM or a mean/median value should be a component of the main figures, as it helps provide context to the timing virus acquisition.

A figure panel was added (Fig. 2D in the revised document) showing the longitudinal median plasma levels (+/- range) of antibody in animals from the fully and partially neutralizing antibody study. (As noted below, these groups were combined in an ad hoc on the basis of the presence of ITS103.01; they were renamed to explicitly acknowledge this.)

4. The use of person-first and de-stigmatizing language is generally excellent throughout the manuscript. The authors should consider the term “at-risk of HIV acquisition” as DAIDs and community groups have identified this as less ideal.

We thank the reviewer for identifying inappropriate language in reference to communities affected by HIV. Corrections were made on line 32 and 45.

5. The viral and mAb features in Figure 2 may be better characterized with an annotated figure of the median +/- range of VL and mAb level, and smaller call out of the comparison between groups of the features in Figs 2 A-D.

The viral load comparisons in figure 2 are not amenable to being supported by a figure of median VL +/- range; if the longitudinal VL data was used (Fig.1B) it would be skewed depending on the proportion of animals in the group that are infected. On the other hand, the synchronized VLs curves in Fig.1C also cannot be plotted according to the median since synchronization results in all the horizontal axis coordinates (timepoints) being shifted according to each animals interpolated “time to 1000 copies/mL”.

We added median +/- range data for the mAb levels in figure 2 (panel D in revised document).

6. The “no apparent vaccinal effect from infusion of mAb” section seems to discuss a lack of immunologic response to viral blips, more than to the mAb. In addition, the change in autologous neutralizing antibody responses may be virus-specific, and far more measurable against the relatively sensitive SIVsmE660 vs. SIVmac239. Autologous nAbs against the latter virus are often delayed or do not occur during the timeframe in which these responses almost universally arise in PWH. Similarly, the impact of the viruses on vaccinal effect should be mentioned in discussion lines 309-310.

We re-worded the results section heading to more clearly encompass both infused mAb-enhanced and viral blip-induced immune response investigations (line 201). We agree that neutralizing responses against SIVmac239 would not normally be expected within the 16-week post-infection timeframe, and we more clearly indicated that we were assaying for Env-binding antibody responses rather than neutralizing responses (line 202). We also added text to the discussion about the difference in viruses employed in therapeutic bNAb studies looking at vaccinal effect versus those used here (i.e. SHIV vs SIV) (line 323-325).

7. The sentence at line 264-266 is ambiguous and should be clarified.

We thank the reviewer for pointing out the confusing line. It has been edited to more clearly explain that the median mAb level of 98-fold over the IC80 where subclinical infections started to occur is comparable to the 200-fold IC80 VRC01 mAb level associated with 90% protective efficacy (line 277-278).

Referee #3 (Remarks to the Author):

Gonelli et al studied breakthrough SIV during the waning phase after administration of neutralising antibodies in macaques. They nicely show with twice weekly monitoring that viral blips are commonly seen prior to full infection. In the instance where the blip could be genotyped it corresponded to a barcoded SIV strain administered 1-2 weeks prior to the blip. The work is well done, has implications to Nab prevention of HIV in humans and should be of interest to Nature readers.

Comments

The viral blips are key to the interpretation of the work. In humans, sensitive viral load assays have been reported to be occasionally be weakly positive in the absence of HIV infection although the veracity of such claims is admittedly unclear. It is also very curious and somewhat worrying that in the second study 4 of the 5 blips detected were at the same time point (day 35) – this seems beyond the boundaries of a statistical fluke? The authors may wish to consider rigorously validating the blips by (a) doing VL tests on a large number of monkeys not exposed to SIV to show that the ~10 blips of ~200 negative VL samples are not spurious, (b) repeating the assays of the blips to show they are reproducible (c) trying harder to sequence the small amount of RNA in the blips - it is worrying that only one sample could be sequenced – in HIV it is common to get a positive sequencing result when the VL approaches 10e3, (d) looking for virus in CD4 T cell DNA (and sequencing if found).

While the viral blips do contribute to indication of subclinical infections, they are not the major form of evidence for suggesting infections occur while mAb levels are high. We show multiple examples (5/6 animals) of unequivocal sequence-based detection at later timepoints of earlier challenged sequence variants, when bNAb levels have declined and when prior challenges are the only potential source of those sequences. The occurrence of 4/5 blips on day 35 in this study could be explained by variability in the challenge (e.g. small change in dilution) that might allow for blips of one lineage to be enhanced; or it is indicative of the reproducibility of the antibody-based blocking. In addition to the two studies shown here, the “blipping” phenomenon has been seen in a single animal in other studies (including a vaccine study of ours), and in two animals in a followup study initiated but not complete for a year still. Together, these studies show that the “blip” infections are real infections; we are as yet unsure of how often they manifest as fulminant infections at this time.

Regarding the points for validating the blips:

- (a) We conducted testing of longitudinal plasma samples from four uninfected rhesus macaques, included as a control cohort in assay runs for a larger cohort of infected animals from an antiretroviral drug study. We tested a total of 88 samples, 22 from each of the 4 animals, sampled over a period of approximately 16 months, and all were scored as negative/below threshold, with a threshold of <15 copies/mL (n=64) or <2 copies/mL (n=24), depending on the assay format.
- (b) Due to the relatively small size of the macaques in the study and frequent sampling, volumes of plasma were limited and did not allow for extensive confirmatory testing. After initial viral load testing and allocation of plasma for attempting barcode sequencing, the remaining volume was sub-optimal for conducting repeat viral load testing with the same sensitivity as the initial test.
- (c) Multiple attempts were made at sequencing the RNA in the blips to determine the barcodes, but as mentioned above the volume of sample available from small animals being frequently bled was a limiting factor. Also, the viral loads for the blips were typically below the level of 1000 copies/mL, and with limited input volumes, likely accounted for the technical failure to obtain sequence data.
- (d) The small sample volumes also limited the number of PBMCs stored from each timepoint. Given the low viral loads observed at the viral blips and our past experience with isolating proviral DNA, it is unlikely we would reliably isolate sequences from these animals. Instead, the

low copy barcodes seen after breakthrough infection that could only have come from challenges 1-8 weeks prior is compelling sequence evidence given the labile RNA could only have come from replicating virus that arose from cells infected while mAb levels were high.

Was the “full” vs “partial” neutralization comparison in Fig 1 and 2 a pre-specified endpoint with an analysis plan or a somewhat ad hoc analysis based on the data – if the latter it would be better to analyse each of the 5 groups separately. Fig 1E is not needed.

The grouped analysis of animals receiving ITS103.01 or not was not a pre-specified endpoint. We clarified in the text that was an ad hoc analysis and adjusted the description of the grouping as “+ ITS103.01 mAb” or “– ITS103.01 mAb” to make clear the distinction between animals is whether the completely neutralizing antibody ITS103.01 was given or not (since not all animals that received ITS103.01 were co-administered partially/incompletely neutralizing mAb) (line 94-96).

In Fig 1B, there seem to be more treated animals that ended up with low set point VLs (one or 2 per group). Was this driven by MHC genetics?

We added information on the MHC genetics for the animals used in the Figure 1 study, specifically whether they encode protective alleles *Mamu-A*01*, *Mamu-B*08*, or *Mamu-B*17* (see supplementary table 5). There are 7 animals with at least one of these alleles and 5 of them tend to lower VLs over time. The other 6 animals with comparable VL setpoints do not have protective MHC alleles. We have updated figure 1B to indicate animal IDs for curves showing a lower setpoint and also marked those with protective alleles (to minimize the need to cross-reference the supplementary table). Given not all animals with protective alleles have reduced VLs and each group has animals without known protective alleles demonstrating similar or lower setpoint viral loads, there are likely other factors contributing to the VLs over time.

Although this work is highly novel, a similar anecdotal phenomenon was described in pigtailed macaques exposed to cell-associated SHIV a few years back with speculation on its implications in a follow up perspective doi: 10.1016/j.it.2017.12.006 doi: 10.1126/scitranslmed.aaf1483. That work speculated that suppressed infections occurring during Bnab treatment could recrudescence later. In this work, the continual SIV challenges made that hard to assess. An interesting follow up study would of course be to stop challenging once a blip is observed.

We thank the reviewer for highlighting these examples from the literature. The cell-associated virus persisting for 7-weeks under PGT121 presence is indeed very similar to the suggested mechanism for where the subclinical infection-associated barcode virus eventually arises from once mAb levels decline. Discussion of this study has been added at line 272.

As we mentioned at line 313, determining whether these subclinical infections behave as typical acute infections (if challenges are halted before mAb levels are low enough for breakthrough infection) is indeed an interesting question and we are considering appropriate designs of experiments to address this point.

Truvada is widely used as pre-exposure prophylaxis and there have been reports of breakthrough HIV infections with some similarities to the blips reported here. Longer acting injectible antiretrovirals are now also being used for Prep. I think this work has potential implications for that field.

We would certainly like to add discussion on this point; however, we have not been able to find references describing viral blips in uninfected individuals taking Truvada. If the reviewer can provide a citation, we would happily add this.